# Single-molecule imaging of telomerase reverse transcriptase in human telomerase holoenzyme and minimal RNP complexes

Robert Alexander Wu[1], Yavuz S Dagdas[2], S Tunc Yilmaz[3], Ahmet Yildiz[1,3]*, Kathleen Collins[1]*

[1]Department of Molecular and Cell Biology, University of California, Berkeley, Berkeley, United States; [2]Biophysics Graduate Group, University of California, Berkeley, Berkeley, United States; [3]Department of Physics, University of California, Berkeley, Berkeley, United States

**Abstract** Telomerase synthesizes chromosome-capping telomeric repeats using an active site in telomerase reverse transcriptase (TERT) and an integral RNA subunit template. The fundamental question of whether human telomerase catalytic activity requires cooperation across two TERT subunits remains under debate. In this study, we describe new approaches of subunit labeling for single-molecule imaging, applied to determine the TERT content of complexes assembled in cells or cell extract. Surprisingly, telomerase reconstitutions yielded heterogeneous DNA-bound TERT monomer and dimer complexes in relative amounts that varied with assembly and purification method. Among the complexes, cellular holoenzyme and minimal recombinant enzyme monomeric for TERT had catalytic activity. Dimerization was suppressed by removing a TERT domain linker with atypical sequence bias, which did not inhibit cellular or minimal enzyme assembly or activity. Overall, this work defines human telomerase DNA binding and synthesis properties at single-molecule level and establishes conserved telomerase subunit architecture from single-celled organisms to humans.

*For correspondence: yildiz@
berkeley.edu (AY); kcollins@
berkeley.edu (KC)

**Competing interests:** The authors declare that no competing interests exist.

## Introduction

Threats to genomic integrity occur at the ends of every linear chromosome, including incomplete DNA synthesis by the replisome and the potential for inappropriate DNA break repair. Eukaryotic cells control these reactions through the function of telomeres, typically consisting of telomeric repeat DNA bound by proteins that comprise the telosome, or in mammalian cells, shelterin (*Palm and de Lange, 2008*; *Stewart et al., 2012*). Telomeric repeat tract maintenance depends on the specialized reverse transcriptase telomerase, which can extend a chromosome 3′ end by processive addition of single-stranded repeats (*Blackburn et al., 2006*). Insufficient telomere synthesis ultimately compromises telomere function and signals a halt to cell proliferation (*O'Sullivan and Karlseder, 2010*; *Aubert, 2014*). This telomere-linked restriction of cellular renewal leads to failures of highly proliferative human tissues, with clinical manifestations including bone marrow failure, aplastic anemia, and pulmonary fibrosis (*Armanios and Blackburn, 2012*).

The active human telomerase ribonucleoprotein (RNP) includes telomerase reverse transcriptase (TERT), which provides the active site, and an RNA (hTR) containing a reiteratively copied internal template. The unique repeat addition processivity of telomerase requires conserved domains in both TERT and hTR that distinguish telomerases from other polymerase families (*Blackburn and Collins, 2011*; *Podlevsky and Chen, 2012*). The TERT N-terminal (TEN) domain allows retention of single-stranded DNA during the template repositioning required for tandem repeat synthesis. TEN-domain-truncated TERT, designated 'TERT ring' based on *Tribolium* TERT structure (*Gillis et al., 2008*), supports only

**eLife digest** Enzymes carry out the many diverse chemical reactions that support life. Some enzymes are made of just one component protein that works on its own, but others are made of multiple proteins that are all required for the enzyme to work properly. Most of what is understood about the activities of enzymes has been deduced by studying solutions containing many enzyme molecules. However, many enzymes can bind to different combinations of proteins to form groups (or 'complexes') with a variety of three-dimensional shapes, so there may be a variety of enzyme complexes in the solution. This can lead to researchers drawing different conclusions about the same enzyme.

In humans and other eukaryotic organisms, DNA is contained within structures called chromosomes. An enzyme called telomerase adds structures called telomeres to the ends of the chromosomes, which protect the DNA from damage. The center of telomerase has a protein called TERT that forms complexes with other proteins. However, it is not known how many copies of the TERT protein are present in each complex. Wu et al. studied these complexes using fluorescent tags that enabled each protein to be identified using a technique called 'single-particle imaging'. The experiments show that these complexes can contain either one or two TERT proteins.

It had previously been suggested that TERT is only an active enzyme when it is bound to another TERT molecule, but Wu et al. show that even complexes with a single TERT are able to add telomeres to DNA. Further experiments used a mutant form of the TERT protein that cannot interact with other TERT molecules and found that complexes that contain this mutant protein still have normal enzyme activity.

Large quantities of purified proteins were used in this study. Therefore, a future challenge will be to refine the method to allow experiments to use much less protein, which would more closely reflect how telomerase is produced in cells.

single-repeat synthesis that can be complemented to high repeat addition processivity by the TEN domain as a separate polypeptide (*Robart and Collins, 2011*; *Wu and Collins, 2014a*). In addition to these and other catalytic activity requirements for TERT and hTR, a biologically functional human telomerase holoenzyme contains two sets of H/ACA proteins (dyskerin, NHP2, NOP10, and GAR1) bound to hTR to direct RNP biogenesis and TCAB1 to redistribute the RNP from nucleoli to Cajal bodies (*Egan and Collins, 2012a*; *Podlevsky and Chen, 2012*; *Schmidt and Cech, 2015*). Telomerase holoenzyme must also assemble with the shelterin protein TPP1 for telomere recruitment and extension of chromosome ends (*Lue et al., 2013*; *Nandakumar and Cech, 2013*; *Sexton et al., 2014*).

Endogenous human telomerase is scarce, with the number of TERT-hTR complexes per cell estimated as only ~35 (*Cohen et al., 2007*) or ~250 (*Xi and Cech, 2014*) in even the most highly telomerase-positive tumor cell lines. Consequently, biochemical investigations of human telomerase have been greatly facilitated by enzyme reconstitution. Enzyme reconstitution in cells exploits transiently introduced plasmids to overexpress TERT and the 451-nucleotide mature hTR, which must be 3′-processed from an appropriate precursor (*Mitchell et al., 1999*; *Fu and Collins, 2003*). Telomerase complexes reconstituted in cells have a diversity of substoichiometric-associated factors (*Egan and Collins, 2012a*; *Nandakumar and Cech, 2013*; *Schmidt and Cech, 2015*). As an alternative reconstitution approach, a minimal-subunit catalytically active RNP can be assembled by expressing TERT in rabbit reticulocyte lysate (RRL) with in vitro transcribed full-length hTR (*Weinrich et al., 1997*) or a half-sized RNA such as hTRmin used here (*Wu and Collins, 2014a*), which lacks the two-hairpin H/ACA motif that assembles the holoenzyme subunits dyskerin, NHP2, NOP10, GAR1, and TCAB1. Only two hTR domains are critical for telomerase catalytic activity: a domain containing the template and adjacent pseudoknot and a branched stem-junction domain containing stem-loop P6.1 (*Mitchell and Collins, 2000*; *Chen et al., 2002*). Importantly, human telomerase enzymes reconstituted in cells or in RRL can interact with the same length of single-stranded DNA, have similar specific activity, and have only minor differences in other enzyme properties such as repeat addition processivity (*Jurczyluk et al., 2010*; *Zaug et al., 2013*; *Wu and Collins, 2014a*).

Central to defining telomerase RNP architecture is a delineation of the number of TERT and hTR subunits that assemble together to generate an enzyme active site. RNP affinity purification and

structural studies indicate a single RNA and single TERT per biologically functional telomerase holoenzyme of single-celled eukaryotes (*Livengood et al., 2002*; *Witkin and Collins, 2004*; *Cunningham and Collins, 2005*; *Hong et al., 2013*; *Jiang et al., 2013*; *Bajon et al., 2015*). This subunit stoichiometry is recapitulated by the minimal *Tetrahymena* telomerase RNP assembled in RRL (*Bryan et al., 2003*). However, the subunit stoichiometry of an active human telomerase RNP is unresolved: some assays suggest TERT and hTR function as monomeric subunits, without dominant-negative inhibition of a wild-type (WT) subunit by co-expressed mutant subunit (*Errington et al., 2008*; *Egan and Collins, 2010*), while other assays suggest obligate co-dependence of active site function across TERT and hTR subunits (*Wenz et al., 2001*; *Sauerwald et al., 2013*). Size fractionation of human telomerase holoenzyme has been suggested to establish TERT dimerization based on molecular mass by gel filtration of ∼600 kDa (*Wenz et al., 2001*) or by glycerol gradient sedimentation of 550 kDa (*Schnapp et al., 1998*) or 670 kDa (*Cohen et al., 2007*) relative to protein standards, but similar fractionation would be predicted for a holoenzyme with a single TERT, single hTR, single TCAB1, and a complex of dyskerin, NHP2, NOP10, and GAR1 bound to each of two H/ACA-motif hairpin stems (*Egan and Collins, 2012a*). Analysis using single-molecule fluorescence correlation spectroscopy detected one TERT and one hTR per RRL-reconstituted minimal RNP (*Alves et al., 2008*). On the other hand, cellular subunit overexpression, purification, and crosslinking yielded particles observed by electron microscopy that were proposed to be active dimeric TERT RNPs, based on detection of two bound single-stranded DNAs (*Sauerwald et al., 2013*). Unfortunately, all of the experiments above suffer from the caveat that individual complexes are inferred to have the activity measured only for a bulk population.

Single-molecule fluorescence microscopy can detect the number of subunits in individual macromolecular complexes. We therefore developed a single-molecule TERT-labeling strategy to determine the TERT subunit content of human telomerase RNPs assembled and purified using methods typical in previous studies. We exploited the preserved function of N-terminally tagged human TERT to introduce the acyl carrier protein (ACP) tag for covalent labeling by prosthetic group transfer from derivatives of Coenzyme A (CoA). ACP and ACP-based tags are well suited to the applications developed here because they are small, monomeric, and expose the conjugated prosthetic group as a conformationally dynamic extension from the protein surface (*Byers and Gong, 2007*; *Chan and Vogel, 2010*). We applied previously developed tag labeling methods (*Yin et al., 2006*; *Zhou et al., 2007*) to investigate the TERT content of individual complexes from purifications of cellular telomerase holoenzyme reconstituted by assembly in human 293T cells and minimal recombinant RNP reconstituted by assembly in RRL. Surprisingly, different affinity purifications yielded different mixtures of complexes monomeric or variously multimeric for TERT. TERT complexes were also heterogeneous in catalytic activity and DNA-binding properties. Complexes with TERT monomer supported DNA synthesis. Apparently non-productive TERT self-association occurred through a low-complexity region of the protein dispensable for RNP catalytic activity. Overall, these studies support the function of human telomerase holoenzyme and minimal recombinant RNPs with a single subunit of TERT and demonstrate an evolutionarily conserved telomerase subunit architecture.

## Results

### Purification-biased TERT subunit content of DNA-bound complexes

To quantify the TERT subunit content of reconstituted human telomerase complexes, we developed a strategy to label individual TERT molecules with a Cy3 or Cy5 fluorophore. The 8 kDa ACP and MCP tags derive from bacterial proteins that accept covalent transfer of the CoA phosphopantetheinyl (Ppant) group to a serine on the protein surface (*Figure 1A*, left). In endogenous bacterial context, the Ppant group serves as a 20 Å swing-arm tether for subsequent transient attachment of the acyl groups that are the carrier proteins' cargo. For labeling ACP/MCP in vitro, the Ppant group of CoA can be pre-conjugated to diverse labels including Cy3, Cy5, or biotin prior to the prosthetic group transfer reaction, such that the Ppant swing-arm becomes a spacer between the label and the protein (*Belshaw et al., 1999*; *George et al., 2004*; *Yin et al., 2006*). ACP synthase catalyzes transfer from a derivatized CoA to the ACP tag but does not label the MCP tag, while either tag is labeled by SFP synthase (*Zhou et al., 2007*). Labeling of a tagged fusion protein by SFP synthase in vitro occurred with >80% efficiency (*Yin et al., 2005*).

Human TERT tagged at the N-terminus supports telomere elongation, whereas telomerase assembled with C-terminally tagged TERT does not (*Counter et al., 1998*; *Wong and Collins, 2006*).

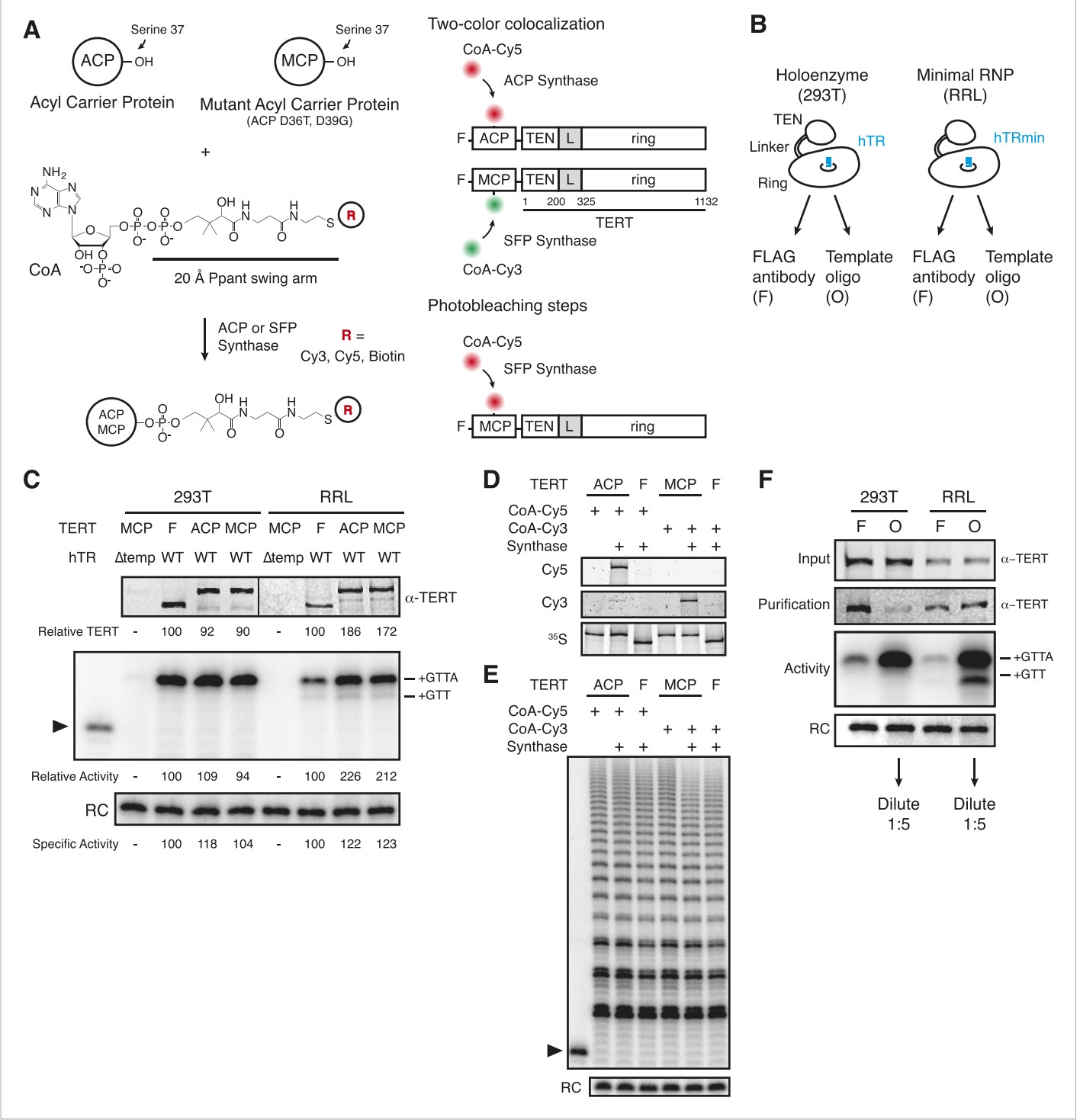

**Figure 1.** Reconstitution, purification, and labeling of human TERT. (**A**) *Left*: derivatized CoA Ppant prosthetic group transfer to acyl carrier protein (ACP) or MCP tag by ACP or SFP synthase. The MCP tag is a modified version of the ACP tag, containing two amino acid substitutions, D36T and D39G. CoA can be modified with dye or biotin groups (R) for enzymatic labeling of a fusion protein. *Right*: schematic of two ACP- and/or MCP-telomerase reverse transcriptase (TERT)-labeling strategies using Cy5 (red) and Cy3 (green). An ACP or MCP tag is N-terminal to the TERT TEN domain, which is connected to the TERT ring by a linker region (L). Numbering refers to the full-length TERT amino acid sequence. A 3xFLAG tag is N-terminal to the ACP or MCP tag. (**B**) Schematic of telomerase holoenzyme reconstitution by overexpression of TERT with full-length hTR in cells (293T) or minimal ribonucleoprotein (RNP) reconstitution by TERT expression with hTRmin in vitro (rabbit reticulocyte lysate [RRL]) followed by FLAG antibody purification for the TEN tag (FLAG antibody purification, F) or purification using a 2'OMe RNA oligonucleotide complementary to the hTR template (Template oligo purification, O). Only the

*Figure 1. continued on next page*

Figure 1. Continued

template of hTR or hTRmin is illustrated (blue). (**C**) TERT and telomerase activity measured for O-purified, eluted complexes. Various N-terminally tagged TERT proteins were detected by TERT antibody immunoblot. The hTR Δtemp reconstitutions used template-less hTR or hTRmin with a 5′ end at hTR position 64. Elution fractions were assayed for telomerase activity by primer extension with dTTP, ddATP, and α-$^{32}$P dGTP, followed by denaturing gel electrophoresis. End-radiolabeled oligonucleotide was added prior to product precipitation to serve as a recovery control (RC), here and in subsequent panels. End-radiolabeled primer is a size marker (▸), here and in subsequent panels. Specific activity in this panel indicates product DNA normalized to amount of TERT. (**D**) SDS-PAGE analysis of RRL-expressed TERT in telomerase reconstitutions of ACP-, MCP-, or only F-TERT in the presence of hTRmin, labeled with $^{35}$S-methionine and any additional label as indicated. ACP synthase was used for ACP-TERT dye labeling and SFP synthase was used for MCP-TERT dye labeling. (**E**) Activity of telomerase reconstituted with ACP-, MCP-, or F-TERT in RRL with hTRmin and labeled as indicated. Activity was detected in reactions containing dATP, dGTP, dTTP, and α-$^{32}$P dGTP, followed by denaturing gel electrophoresis. (**F**) TERT content and telomerase activity in bulk purifications of MCP-TERT reconstituted in 293T cells or RRL, assayed as described in (**C**). TERT immunoblot with input extracts used 3% of the total purification input. Half of the post-purification sample was used for activity assays and half for TERT immunoblot. For single-molecule detection, O-purifications were diluted relative to F-purifications from the same extract. The following figure supplement is available for *Figure 1—figure supplement 1*.

The following figure supplement is available for figure 1:

**Figure supplement 1**. Methods of human telomerase reconstitution and purification.

Therefore, we fused the human TERT N-terminus to a triple FLAG peptide and either the ACP or MCP tag (*Figure 1A*, right). To assemble telomerase holoenzyme, TERT was overexpressed in 293T cells along with full-length hTR overexpressed using the U3 small nucleolar RNA promoter (*Fu and Collins, 2003*). To reconstitute catalytically active minimal RNP, TERT was expressed in RRL pre-supplemented with vast molar excess of purified recombinant hTRmin (*Wu and Collins, 2014a*). TERT complexes from each reconstitution method were enriched by each of two purification approaches: TERT binding to FLAG antibody resin followed by peptide elution (F purification) or RNA template base-pairing to a resin-immobilized 2′O-methyl RNA (2′OMe) oligonucleotide followed by displacement oligonucleotide elution (O purification; *Figure 1—figure supplement 1*, panels A, B; *Schnapp et al., 1998*). The 3′-modified displacement oligonucleotide used in this work did not compete with DNA primer for telomerase elongation (*Figure 1—figure supplement 1*, panel C). 293T cell lysates or RRL expression reactions were split and purified in parallel using the F and O purification approaches (*Figure 1B*). ACP- and MCP-tagged TERTs expressed at equivalent level and assembled active telomerase, quantified by radiolabeled dGTP incorporation in reactions also containing dTTP and ddATP (*Figure 1C*). CoA-Cy5 labeling of ACP-TERT using ACP synthase and CoA-Cy3 labeling of MCP-TERT using SFP synthase were confirmed by SDS-PAGE and fluorescence scanning, with no labeling of TERT lacking an ACP or MCP tag (*Figure 1D*). Importantly, the profile of telomerase product synthesis was not affected by the labeling reaction for fluorophore conjugation (*Figure 1E*).

While purification by either tagged TERT or RNA template yields active telomerase, these purification strategies also enrich either TERT not assembled with hTR or hTR without TERT, respectively. To investigate the amounts of tagged TERT vs active RNP, we compared the levels of TERT protein and enzyme activity across the four combinations of reconstitution and purification, subsequently designated 293T-F, 293T-O, RRL-F, and RRL-O (*Figure 1B,F*). TERT was detected by an antibody raised against its C-terminal region (*Figure 1—figure supplement 1*, panel D). Activity was quantified from reactions with dTTP, ddATP, and radiolabeled dGTP. O-purification by the hTR template enriched more telomerase activity relative to TERT than did F-purification (*Figure 1F*), as would be expected based on template hybridization vs antibody binding to TERT. Comparison between the pair of 293T or RRL purifications suggests that most of the TERT in 293T-F and RRL-F was not assembled as telomerase RNP. This was anticipated for the 293T-F purification, because cellular expression of hTR is limited by inefficient co-transcriptional H/ACA RNP assembly (*Darzacq et al., 2006*; *Egan and Collins, 2012b*). However, RRL reconstitution exploits the use of pre-transcribed hTR added at very high final concentration relative to TERT. Nonetheless, even optimized RRL expression produced hTR-free TERT enriched by F-purification.

To investigate the TERT content of individual complexes within a bulk fraction, we used total internal reflection fluorescence microscopy to image labeled TERT complexes bound to immobilized single-stranded $T_{15}(T_2AG_3)_2$ DNA primer. This 5′-biotinylated primer was anchored to a polyethylene glycol-coated coverslip surface via biotin–streptavidin attachment (*Figure 2A*, left). Primers with this 3′ permutation of the telomeric repeat have exceptionally stable binding to human telomerase (*Wallweber et al., 2003*) due to the finely tuned recognition of template-paired primer 3′ ends in the

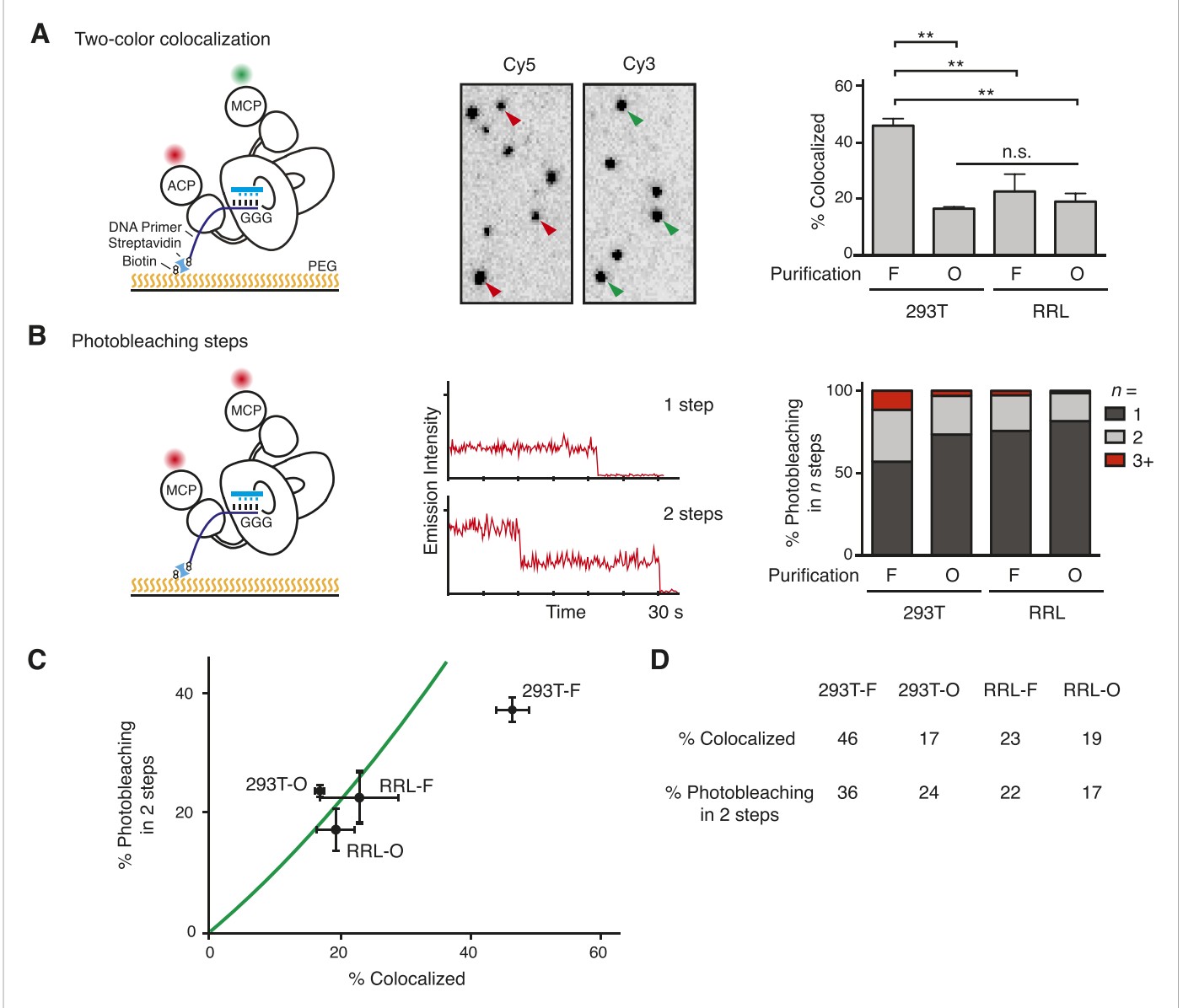

**Figure 2**. Single-molecule detection of the TERT subunit content in DNA-bound complexes. (**A**) *Left*: schematic for detection of TERT content by two-color co-localization. ACP-TERT was labeled with Cy5 (red) and MCP-TERT was labeled with Cy3 (green). PEG indicates polyethylene glycol. *Center*: example of detection of two-color co-localization indicated by arrowheads, for a 293T-F sample. *Right*: percentage of two-color co-localization for DNA-bound complexes with co-expressed ACP- and MCP-TERTs, purified by the TERT tag (F) or template-complementary 2'OMe RNA oligonucleotide (O). For this and subsequent quantifications, values are averaged from three assays using experimentally independent replicates with standard error of the mean shown. **$p < 0.01$ using one-way ANOVA, followed by Tukey's multiple comparison test; n.s. is not significant. (**B**) *Left*: schematic for detection of TERT content by steps of photobleaching. MCP-TERT was labeled with Cy5 (red). *Center*: examples of photobleaching in one or two steps. *Right*: percentage of MCP-TERT DNA-bound complexes labeled with Cy5 that photobleached in one, two and three, or more (3+) steps. Values are the average of triplicate experimental replicates. (**C**) The predicted relationship between detections of TERT subunit co-localization and two-step photobleaching is shown as the green line (see Materials and methods, Equation 3). Data were plotted according to measured co-localization and photobleaching in two steps only. Error bars represent standard error of the mean from triplicate experimental replicates of each measured parameter. (**D**) Measured two-color co-localization and two-step photobleaching as determined by the experiments in (**A**) and (**B**), respectively. The following figure supplement is available for *Figure 2—figure supplement 1*.

The following figure supplement is available for figure 2:

**Figure supplement 1**. Technical robustness of the two-color co-localization assay for TERT subunit content.

enzyme active site (*Brown et al., 2014*; *Wu and Collins, 2014b*). Optimal DNA binding by human telomerase requires a primer length of two telomeric repeats (*Wallweber et al., 2003*), which is the same length that active human telomerase protects from nuclease digestion (*Wu and Collins, 2014a*).

We applied two parallel approaches to determine the number of TERT subunits per complex. In the first method (*Figure 2A*, left), we assembled telomerase by co-expression of ACP-TERT and MCP-TERT and labeled the TERTs sequentially, first labeling ACP-TERT with ACP synthase and CoA-Cy5 then labeling MCP-TERT with SFP synthase and CoA-Cy3. Fields of individual complexes were imaged to detect both dyes, and images were scored for the fraction of Cy5-labeled ACP-TERT that co-localized a Cy3-labeled MCP-TERT. In the second TERT subunit counting method (*Figure 2B*, left), we assembled telomerase complexes containing only MCP-TERT and labeled using SFP synthase and CoA-Cy5. Fields of individual complexes were imaged, and each Cy5 'spot' in the flow cell was analyzed for the number of dye photobleaching steps that occurred before the spot vanished. In the parallel approaches, the fraction of two-color co-localized spots and the number of photobleaching steps are both readily related to the sample fractional content of TERT monomer and dimer considering all possible two-subunit combinations (see Materials and methods, Equations 1, 2). Fluorescently labeled TERT complexes were diluted to obtain 1–4 spots per 100 $\mu m^2$ of the slide surface, and unbound protein was removed before imaging. To attain similar spot count per field across samples, labeled O-purification complexes required dilution relative to F-purification complexes isolated from an equal amount of the same extract, consistent with the greater yield of active RNP for O-purification (*Figure 1F*).

Both two-color co-localization and photobleaching assays revealed the presence of more than one labeled TERT in a subset of the DNA-bound TERT complexes (*Figure 2A,B*). In the two-color co-localization assay, there was no statistically significant difference in TERT co-localization comparing DNA-bound 293T-O, RRL-F, and RRL-O complexes (*Figure 2A*; 17–23%, p = 0.58). In contrast, 293T-F complexes had much more TERT co-localization (46%, p = 0.0015). This distinction was consistent across a range of fluorescent spot density per field and different 293T cell extracts used for purifications (*Figure 2—figure supplement 1*). Results from the photobleaching method of TERT subunit counting also indicated no statistically significant difference in TERT subunit content across the population of DNA-bound complexes from 293T-O, RRL-F, and RRL-O (*Figure 2B*; 19–27% bleaching in multiple steps, p = 0.33). In contrast, 293T-F complexes had much more multistep photobleaching (43%, p = 0.0066) including a substantial fraction of complexes that photobleached in three or even more than three steps (12%). We analyzed whether the results from the methods of subunit counting were consistent with each other, assuming a mixed population of TERT monomer and TERT dimer complexes (Materials and methods, Equation 3, and see below). There is excellent correlation of two-color co-localization to two-step photobleaching results for 293T-O, RRL-F, and RRL-O but not 293T-F (*Figure 2C*). The abundance of 293T-F TERT complexes that photobleached in three or more steps is likely responsible for this discord, as this population of complexes was distinguished from TERT dimer complexes in the count of photobleaching steps but would be lumped together with TERT dimer complexes in the count of two-color co-localization. Together, the findings above reveal a surprising diversity of TERT subunit content in DNA-bound complexes. Furthermore, it is evident that this heterogeneity varies across the methods of telomerase reconstitution and purification (*Figure 2D*).

## Quantification of the TERT monomer fraction of DNA-bound complexes

The subunit co-localization and multistep photobleaching values measured above are related to the number of TERTs within each complex but are also influenced by the labeling efficiency of the MCP tag. Therefore, in order to calculate the fraction of complexes with TERT monomer or TERT dimer for each method of reconstitution and purification, it was necessary to establish MCP-TERT-labeling efficiency. Labeling of 293T- and RRL-reconstituted, F-purified MCP-TERT complexes was to saturation within 30 min of a reaction with CoA-Cy3 or CoA-Cy5 (*Figure 3A*), well within the standard 2-hr labeling protocol. Also, labeling efficiency was not dependent on the reconstitution and purification method (*Figure 3B*). We adapted a previously developed approach to quantify a minimum lower bound of labeling efficiency without assumptions from fluorescence intensity (*Yin et al., 2005*). Using CoA-biotin as the synthase substrate results in covalent target protein biotinylation, which can be used as the basis for protein depletion by binding to streptavidin resin. The minimum lower bound of labeling efficiency can be calculated from the amount of protein remaining in the unbound fraction. First, we confirmed that CoA-biotin is used equivalently to CoA-fluorophore by measuring competition between CoA-biotin

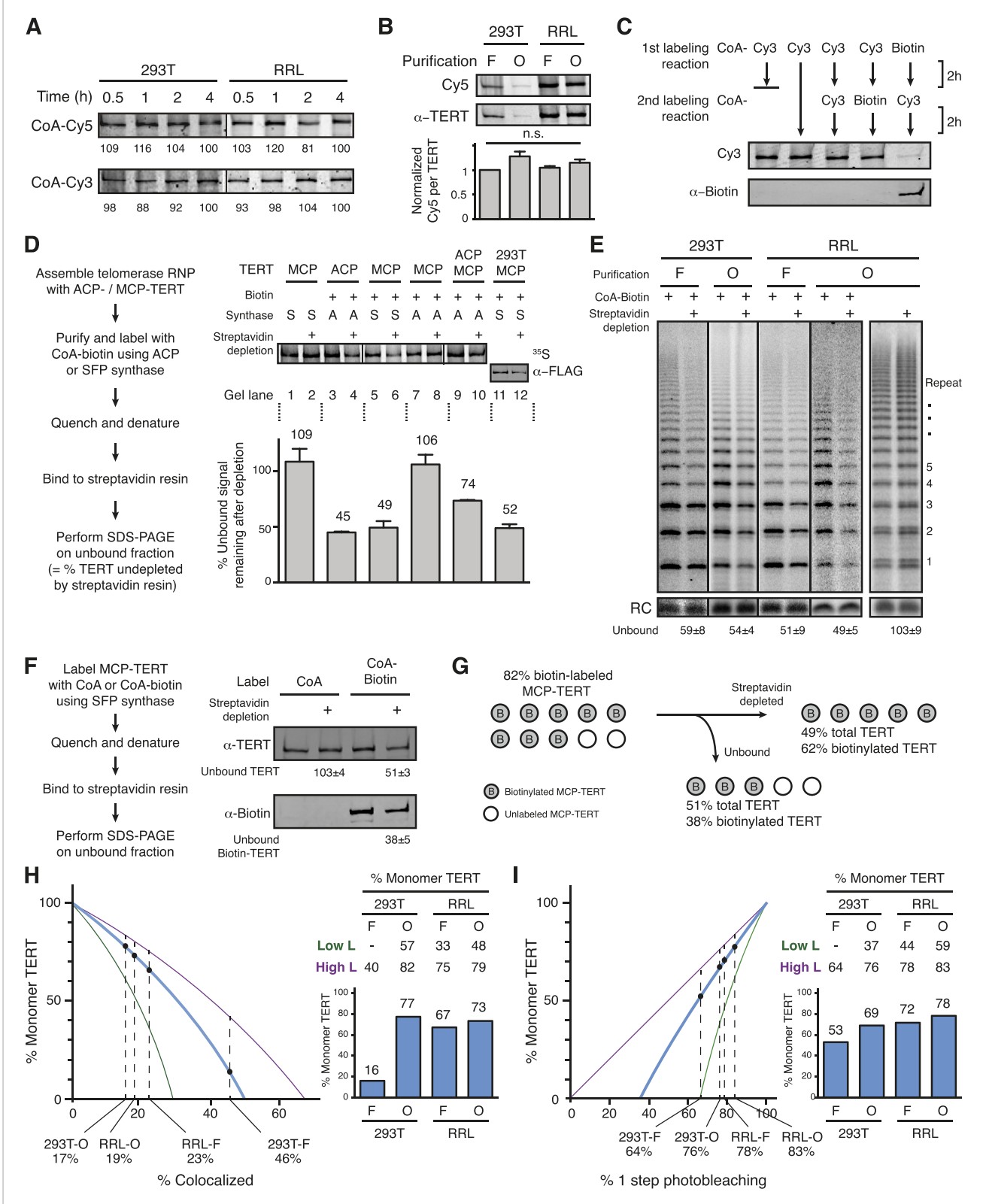

**Figure 3**. Quantification of the TERT monomer vs multimer content in purified samples based on TERT-labeling efficiency. (**A**) SDS-PAGE analysis of the kinetics of labeling F-purified 293T- or RRL-reconstituted MCP-TERT in reactions with CoA-Cy5 or CoA-Cy3 and SFP synthase. Lines within the panel indicate separate sets of gel lanes. Quantification of labeling intensity was normalized to labeling at the 4-hr time point after subtraction of background.

*Figure 3. continued on next page*

Figure 3. Continued

(**B**) Cy5 labeling relative to TERT amount analyzed for telomerase reconstituted and purified as indicated. TERT was detected by TERT immunoblot. Values are the average of triplicate experimental replicates. (**C**) Validation of equivalent labeling using CoA-dye or CoA-biotin by sequential labeling of F-purified, RRL-expressed MCP-TERT with SFP synthase. Initial TERT labeling using CoA-Cy3 or CoA-biotin competes for subsequent TERT labeling by the other CoA derivative. The biotin label on MCP-TERT was detected by biotin antibody immunoblot. (**D**) *Left*: schematic of the biotinylated TERT depletion procedure. *Right*: quantification of ACP- and/or MCP-TERT remaining after streptavidin agarose depletion, following reconstitution (RRL unless indicated otherwise), F-purification and labeling using CoA-biotin and ACP (A) or SFP (S) synthase. RRL-expressed TERT was $^{35}$S-methionine labeled and 293T-expressed TERT was detected by FLAG antibody immunoblot. Samples labeled in reactions lacking CoA-biotin (not Biotin +) were labeled with CoA and those not applied to streptavidin agarose (not Streptavidin depletion +) were mock-depleted on Myc antibody agarose. Lines within the panel indicate separate sets of gel lanes run in parallel. Percentage unbound was calculated as unbound signal normalized to unbound signal of the control depletion. Values are the average of triplicate experimental replicates. (**E**) Activity of the unbound fraction after streptavidin agarose depletion of biotinylated telomerase labeling using CoA-biotin and SFP synthase, under native binding conditions. Telomerase activity was assayed in reactions with dATP, dGTP, dTTP, and $\alpha$-$^{32}$P dGTP, followed by denaturing gel electrophoresis; number of 6-nucleotide repeats added to product DNA is indicated. Samples not depleted with streptavidin agarose were mock-depleted on Myc antibody agarose. Lines within the panel indicate separate sets of gel lanes run in parallel. Percentage unbound was normalized to unbound after control depletion. Values are the average of triplicate experimental replicates. (**F**) *Left*: schematic of the biotinylated TERT depletion procedure and unbound fraction analysis. *Right*: quantification of total TERT and biotinylated MCP-TERT in the unbound fraction of 293T-reconstituted, F-purified telomerase, following labeling using CoA-biotin or CoA and depletion by streptavidin agarose or mock-depletion on Myc antibody agarose. MCP-TERT and the biotin label on MCP-TERT were detected by immunoblot. Values are the average of triplicate experimental replicates. (**G**) Illustration of labeling efficiency determination by comparison of the percent unbound total MCP-TERT and unbound biotinylated MCP-TERT. (**H**) Calculated percentage of DNA-bound TERT monomer complexes according to fraction TERT subunit co-localization (percentages indicated), assuming the TERT-labeling efficiency measured value (82%, blue line; bar graph at *right*), lower bound (51%, green line; Low L numbers at *right*), or upper bound (100%, purple line; High L numbers at *right*). Vertical dashed lines are the observed fraction of two-color co-localization (from *Figure 2A*). (**I**) Calculated percentage of DNA-bound TERT monomer complexes according to fraction of one-step photobleaching (percentages indicated), assuming the TERT-labeling efficiency measured value (82%, blue line; bar graph at *right*), lower bound (51%, green line; High L numbers at *right*), or upper bound (100%, purple line; Low L numbers at *right*). Vertical dashed lines are the observed fraction of one-step photobleaching (from *Figure 2B*).

and CoA-Cy3. If RRL-reconstituted F-purified MCP-TERT was labeled to saturation with CoA-Cy3 and then labeled with CoA-biotin, no biotinylation of TERT could be detected (*Figure 3C*). Similarly, labeling with CoA-biotin drastically reduced subsequent labeling with CoA-Cy3 (*Figure 3C*). Therefore, biotin labeling provides a surrogate for quantification of dye labeling efficiency.

To preclude the depletion of unlabeled TERT as part of a TERT multimer, biotin-labeled TERT complexes were bound to streptavidin in 2 M urea. Complexes of RRL-reconstituted TERT synthesized with $^{35}$S-methionine were F-purified and labeled with biotin, biotinylated TERT was depleted using streptavidin agarose, and the fraction of unbound TERT was quantified by radiolabel detection after SDS-PAGE (*Figure 3D*). Depletion was an indistinguishable 54.9 ± 0.7% or 50.6 ± 5.9% of ACP-TERT or MCP-TERT, respectively, quantified from the unbound ~45% or ~49% (*Figure 3D*, lanes 3–6). As a negative control, TERT from labeling reactions with underivatized CoA was not depleted (*Figure 3D*, lanes 1–2). Also, MCP-TERT was not depleted after a labeling reaction with ACP synthase (*Figure 3D*, lanes 7–8). When ACP- and MCP-TERT were co-expressed, CoA-biotin labeling of ACP-TERT by ACP synthase resulted in half the depletion attained when ACP-TERT alone was expressed (*Figure 3D*, lanes 9–10 vs 3–4), confirming equal co-expression of the two tagged TERTs. Similar depletion was observed for biotin-labeled F-purified MCP-TERT expressed in 293T cells, detected by immunoblot (*Figure 3D*, lanes 11–12). Also, ~50% depletion was observed for the catalytic activity of MCP-TERT RNPs labeled with CoA-biotin bound to streptavidin in native rather than denaturing conditions, independent of the reconstitution or purification method or enzyme repeat addition processivity (*Figure 3E*).

To measure how efficiently biotinylated TERT was depleted by streptavidin in the 2 M urea condition that converts the entire population of protein to monomer, we determined the fraction of biotinylated TERT that was depleted compared to the fractional depletion of total TERT. For maximal immunoblot detection sensitivity, the 293T-expressed F-purified MCP-TERT was biotin labeled, allowed to bind streptavidin then analyzed by immunoblot with antibodies specific for TERT and biotin (*Figure 3F*). Streptavidin depleted 49% of the TERT protein (*Figure 3F*), consistent with the previous TERT depletions (*Figure 3D*). However, 38% of biotinylated TERT remained unbound (*Figure 3F*), revealing that streptavidin binding in 2 M urea did not completely deplete the labeled TERT. Therefore, the MCP-TERT-labeling efficiency was much higher than 51%. Correcting the

quantified total TERT depletion for depletion efficiency of biotin-TERT gives an MCP-TERT-labeling efficiency of 82% (*Figure 3G*). This matches the labeling efficiency determined for a related tag in similar reactions with SFP synthase and CoA-biotin (*Yin et al., 2005*).

We determined the fraction of TERT monomer vs dimer in each population using the co-localization quantifications (*Figure 2A*, Materials and methods, Equation 1) or the photobleaching quantifications (*Figure 2B*, Materials and methods, Equation 2) by modeling the DNA-bound complexes as having either one or two TERTs. We set labeling efficiency as 82% but also modeled a range of labeling efficiency from 51% to 100% as lower and upper bounds (indicated as 'Low L' and 'High L' limits). Co-localization and photobleaching quantifications support modeling of DNA-bound 293T-O, RRL-F, and RRL-O populations as a mixture of complexes with one or two TERT subunits (*Figure 3H,I*). The 293T-F population of DNA-bound TERT could not be modeled as a mixture of TERT monomer and dimer across the full range of labeling efficiency using the co-localization quantification (*Figure 3H*), likely due to the substantial fraction of complexes with three or more TERT subunits (*Figure 2B*). TERT monomer complexes exceeded TERT dimer complexes in the DNA-bound 293T-O, RRL-F, and RRL-O populations across almost the entire range of modeled labeling efficiencies (*Figure 3H,I*). Overall, these analyses establish that TERT complexes competent for DNA binding can have a single subunit of TERT.

## Assessing the active RNP fraction of DNA-bound TERT complexes

The heterogeneity of TERT subunit content in DNA-bound complexes described above raised the question of whether only a subset of the DNA-bound complexes corresponds to active RNP. To investigate this question, we exploited the permutation-dependent telomeric-repeat DNA-binding affinity of the human telomerase active site. The single-stranded $T_{15}(T_2AG_3)_2$ DNA primer used to bind TERT complexes to the flow cell surface has extremely slow dissociation from the telomerase holoenzyme active site (*Wallweber et al., 2003*). Introducing dTTP + dATP into the imaging chamber would support primer extension to a GGGTTA-3′ end (*Figure 4A*), which disengages from the active site with $k_{off}$ at least ~100-fold greater than the TTAGGG-3′ end (*Wallweber et al., 2003*). Thus, Cy5-labeled MCP-TERT complexes with DNA bound in a functional active site would exhibit activity-dependent elution in buffer with dTTP + dATP (*Figure 4A*). Inactive RNP and hTR-free TERT would remain bound as well as some active RNP not dissociated from product (*Figure 4A*), and also any 293T TERT bound to DNA indirectly through an associated shelterin complex. To control for activity-independent dissociation of TERT complexes from DNA, we performed parallel incubations without dTTP + dATP. We also assayed complexes reconstituted with the catalytic-dead TERT D868A (*Weinrich et al., 1997*). Elution of Cy5-labeled MCP-TERT was monitored by spot count per field over 30 min in buffer with or without dNTPs (*Figure 4B*).

Reproducibly, more elution of WT TERT complexes occurred in the presence of buffer with dTTP + dATP vs buffer alone (*Figure 4C*, compare black and gray). In contrast, D868A TERT complexes showed the same amount of dissociation with or without dNTPs (*Figure 4C*, compare dark and light blue). Curiously, the fraction of WT TERT complexes with activity-dependent elution varied widely across the TERT populations from different reconstitution and purification conditions (*Figure 4C*). RRL-F and RRL-O complexes showed predominantly activity-dependent elution. About half of the DNA-bound 293T-O complexes also showed activity-dependent elution, but a surprisingly low percentage of 293T-F complexes eluted with the opportunity for DNA synthesis. The non-eluting fraction of TERT complexes roughly correlated with the fraction of complexes that could bind slide-immobilized DNA after sample pre-treatment with RNase A (*Figure 4C*, gray bars).

Importantly, the fraction of DNA-bound 293T-O, RRL-F, and RRL-O TERT complexes with nucleotide-dependent-'specific' elution (*Figure 4C*) overlaps the fraction of DNA-bound complexes with monomeric TERT (*Figure 3H,I*). This overlap establishes that at least some TERT monomer RNPs have catalytic activity. To directly measure the contribution of TERT monomer RNPs to specific elution, we used RRL-reconstituted O-purified Cy5-labeled MCP-TERT complexes to quantify TERT spot count per field and steps of photobleaching for samples after incubation in parallel for 30 min in buffer with or without dNTPs (*Figure 4D*). Approximately one third as many labeled TERT complexes were present in samples incubated with dNTPs (*Figure 4E*), consistent with the elution time course (*Figure 4C*). The reduction of TERT spot count by activity-based elution occurred entirely in complexes with single-step photobleaching (*Figure 4E*, p = 0.0006). The conditions of elution altered the relative representation of TERT monomer and dimer complexes, calculated by adjusting the

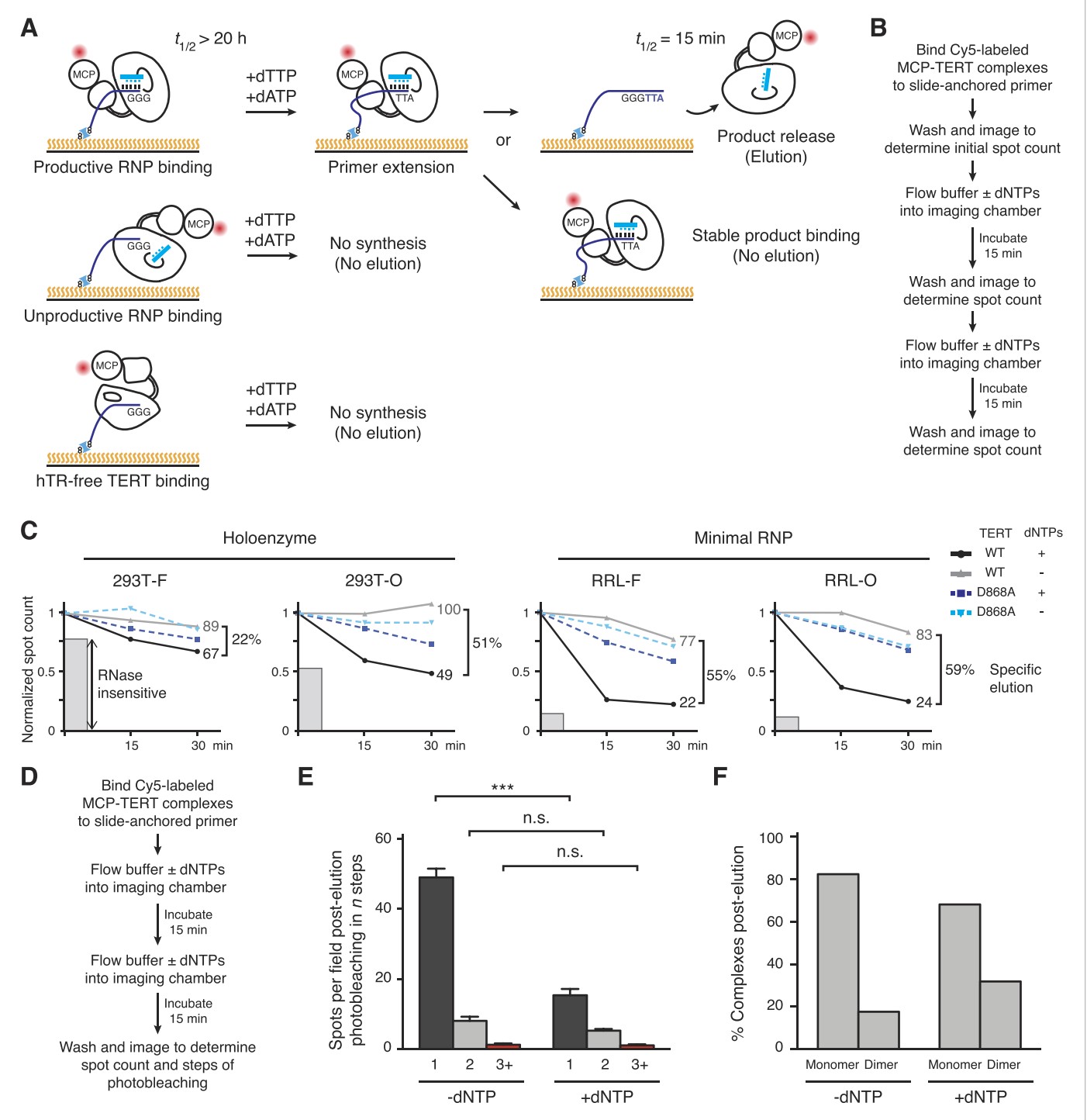

**Figure 4**. Distinct profiles of activity-dependent elution across populations of TERT complexes. (**A**) Schematic of TERT complexes' interaction with bound DNA. In the presence of dTTP and dATP, complexes bound productively to primer end GGG-3' would elongate the primer to GGGTTA-3' accompanied by increased likelihood of DNA release (elution, at *top*). Non-productively bound RNP complexes and hTR-free TERT would not elongate the primer and therefore not elute by DNA synthesis, and some productively bound RNP complexes could also fail to elongate primer and/or to release from product DNA. The $t_{1/2}$ values are from published studies using human telomerase holoenzyme (**Wallweber et al., 2003**). (**B**) Schematic of the activity-dependent elution procedure. (**C**) Activity-dependent elution of Cy5-labeled wild-type (WT) or catalytic-dead (D868A) MCP-TERT complexes using buffer containing dATP + dTTP or buffer only. Spot count per field of labeled TERT complexes was normalized to the initial time point. Specific elution was calculated by subtracting the fraction of complexes with buffer-only elution from the fraction eluted with dNTPs. The relative count of DNA-bound complexes from sample pre-treated with RNase A is indicated by shaded gray bars. (**D**) Schematic of the procedure for post-elution counting and photobleaching of

*Figure 4. continued on next page*

*Figure 4. Continued*

labeled complexes. (**E**) Number of MCP-TERT DNA-bound complexes labeled with Cy5 per imaging field that photobleached in one, two and three, or more steps after elution incubation with or without dNTPs. ***p < 0.001 by unpaired Student's *t*-test, n.s. is not significant. (**F**) Calculated percentage of DNA-bound TERT monomer and dimer complexes after elution according to fractional one-step photobleaching, assuming 82% TERT-labeling efficiency.

photobleaching step quantifications for TERT-labeling efficiency (*Figure 4F*). Consistent with specific elution of TERT monomer complexes, the DNA-bound TERT complexes remaining after specific elution were enriched for TERT dimer. Overall, the findings above strongly suggest that human telomerase catalytic activity requires only a single TERT subunit per RNP.

## Assessing the DNA-binding affinity of TERT complexes

The heterogeneity of DNA-bound TERT complex elution was surprising. We therefore investigated whether the bulk populations of TERT complexes from different reconstitution and purification conditions had heterogeneous DNA-binding affinities as well. Towards this goal, we quantified the DNA-binding affinity of TERT complexes anchored directly to the flow cell surface. To do this, we labeled MCP-TERT with CoA-biotin, bound the biotin-labeled TERT complexes to streptavidin on the flow cell surface, and assayed the immobilized TERT complexes for retention of Cy5-labeled $(T_2AG_3)_2$ (*Figure 5A*). This direct TERT immobilization captured the full TERT heterogeneity of the bulk purification fractions, which we monitored separately by SDS-PAGE of MCP-TERT labeled with Cy5

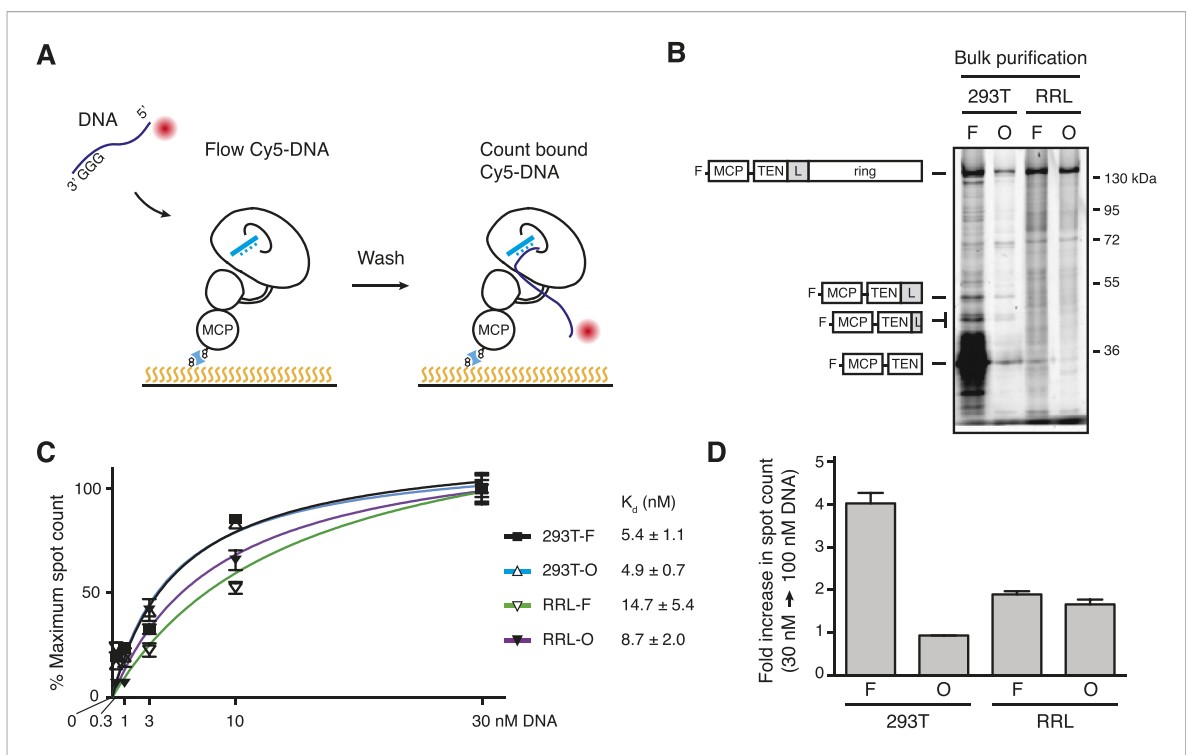

**Figure 5**. Direct DNA-binding affinity comparison for TERT complexes in bulk purifications. (**A**) Schematic for detection of Cy5-labeled DNA binding to biotinylated TERT complexes. (**B**) SDS-PAGE analysis of MCP-TERT complexes labeled using CoA-Cy5. Cy5-labeled MCP-TERT proteolysis products that retain the N-terminal F-MCP tag and are enriched in the 293T-F purification are schematized in comparison to full-length TERT. (**C**) Concentration dependence of Cy5-labeled DNA retention by slide-anchored TERT complexes across a titration of 0.3, 1, 3, 10, and 30 nM DNA. Spot count per field was normalized to the 30 nM DNA quantification for each sample. Error bars represent standard error of the mean of spot counts of five fields per sample per DNA concentration. (**D**) Graph of the change in Cy5-labeled DNA spot count comparing assays of 30 vs 100 nM DNA, normalized to the 30 nM DNA quantifications for each sample. Error bars represent standard error of the mean of spot counts of five fields per sample per DNA concentration.

(*Figure 5B*). Bulk 293T-F purifications of MCP-TERT contained a large amount of a proteolytic product corresponding to the MCP-tagged TERT TEN domain alone (*Figure 5B*). TEN domain expressed in *Escherichia coli* has barely detectable if any DNA-binding activity (*O'Connor et al., 2005*; *Sealey et al., 2010*), suggesting that it would not form a stable complex with the Cy5-labeled $(T_2AG_3)_2$. None of the other bulk purification fractions contained TEN domain alone, but curiously the 293T-F and 293T-O bulk purifications contained TERT proteolytic products corresponding to the TEN domain plus adjacent linker (*Figure 5B*; see below).

We quantified the amount of Cy5-labeled $(T_2AG_3)_2$ bound to immobilized TERT complexes using a range of DNA concentration. DNA binding across a titration from 0.3 to 30 nM DNA yielded $K_d$ calculations in nM of 4.9 ± 0.7 for 293T-O, 5.4 ± 1.1 for 293T-F, 8.7 ± 2.0 for RRL-O, and 14.7 ± 5.4 for RRL-F (*Figure 5C*). The ~5 nM $K_d$ of holoenzyme and ~10 nM $K_d$ of minimal RNP are consistent with the holoenzyme $K_m$ for elongation of similar primers measured, under different conditions, as 2 nM or 8 nM (*Wallweber et al., 2003*; *Jurczyluk et al., 2010*). In parallel, immobilized TERT complexes were assayed for DNA binding using 100 nM $(T_2AG_3)_2$. DNA binding by 293T-F TERT complexes increased ~fourfold with 100 nM compared to 30 nM DNA (*Figure 5D*). In contrast, at 100 nM compared to 30 nM DNA concentration, 293T-O TERT complexes showed no additional DNA binding and RRL-F and RRL-O complexes showed only limited additional association with DNA (*Figure 5D*). These findings suggest that 293T-O, RRL-F, and RRL-O TERT complexes competent for DNA binding have a relatively homogeneous DNA-binding affinity matching the expectation for catalytically active human telomerase.

## A TERT linker region not required for telomerase RNP assembly or activity

Next, we sought to create a homogeneous pool of TERT monomer or dimer complexes. Many variations of reconstitution method had surprisingly little impact on the DNA-bound TERT monomer/dimer ratio, with one exception: elimination of the 125 amino acid linker between the TERT ring and TEN domain. Phylogenetic comparison revealed that this domain linker is particularly long in vertebrate TERTs (*Podlevsky et al., 2008*), approximately 100 amino acids longer than in the ciliate and budding yeast TERTs that assemble only TERT monomer RNPs (*Livengood et al., 2002*; *Bryan et al., 2003*; *Witkin and Collins, 2004*; *Cunningham and Collins, 2005*; *Jiang et al., 2013*; *Bajon et al., 2015*). Scanning six-residue substitutions of human TERT linker sequence did not uncover any significance of the region for telomerase catalytic activity or telomere maintenance (*Armbruster et al., 2001*), but this approach did not alter the atypical amino acid composition of the linker region overall. Human TERT amino acids 201–325 are 18% proline, 14% arginine, and 12% glycine. When subject to bioinformatical analysis for amino acid content (*Harbi et al., 2011*), this region is identified as having high compositional bias. We also used SEG analysis (*Wootton and Federhen, 1993*) to search for low-complexity sequence within the human TERT linker. SEG analysis identified two segments of the linker, residues 213–248 and 313–323, as low complexity. Low-complexity regions can mediate diverse protein–protein interactions including concentration-dependent self-association (*Coletta et al., 2010*; *Kato et al., 2012*). Thus, the TERT low-complexity proline/arginine/glycine-rich linker (termed the PAL) is a candidate region for mediating self-association of overexpressed TERT.

Whether the length of the human TERT PAL influences RNP assembly or activity has not been tested. Previous assays that separated the TEN domain from the TERT ring retained the PAL on either the TEN domain or TERT ring (*Robart and Collins, 2011*; *Wu and Collins, 2014a*). We therefore removed TERT residues 201–325 from the N-terminally F-tagged full-length protein, either by simply deleting the region (TERT-ΔPAL; *Figure 6A*) or replacing it with 5, 10, or 20 repeats of the sequence NAAIRS (TERT-5N, −10N, −20N; *Figure 6A*), the six amino acid motif used previously in the non-disruptive scanning mutagenesis (*Armbruster et al., 2001*). The PAL-mutant TERT proteins expressed at levels similar to WT TERT in 293T cells and in RRL (*Figure 6A*), and binding of WT and PAL-mutant TERT complexes to FLAG antibody resin enriched similar amounts of catalytic activity (*Figure 6B*). Although direct fusion of the TEN domain to the TERT ring did not substantially affect the quantified overall activity it appeared to reduce the amount of the longest product DNAs (*Figure 6B*). This change in product profile was rescued by NAAIRS repeat insertion (*Figure 6B*). Since the number of radiolabeled dGTP nucleotides incorporated into a product DNA is proportional to length, products elongated by many repeats are detected with disproportionately high sensitivity relative to their actual abundance. To more accurately

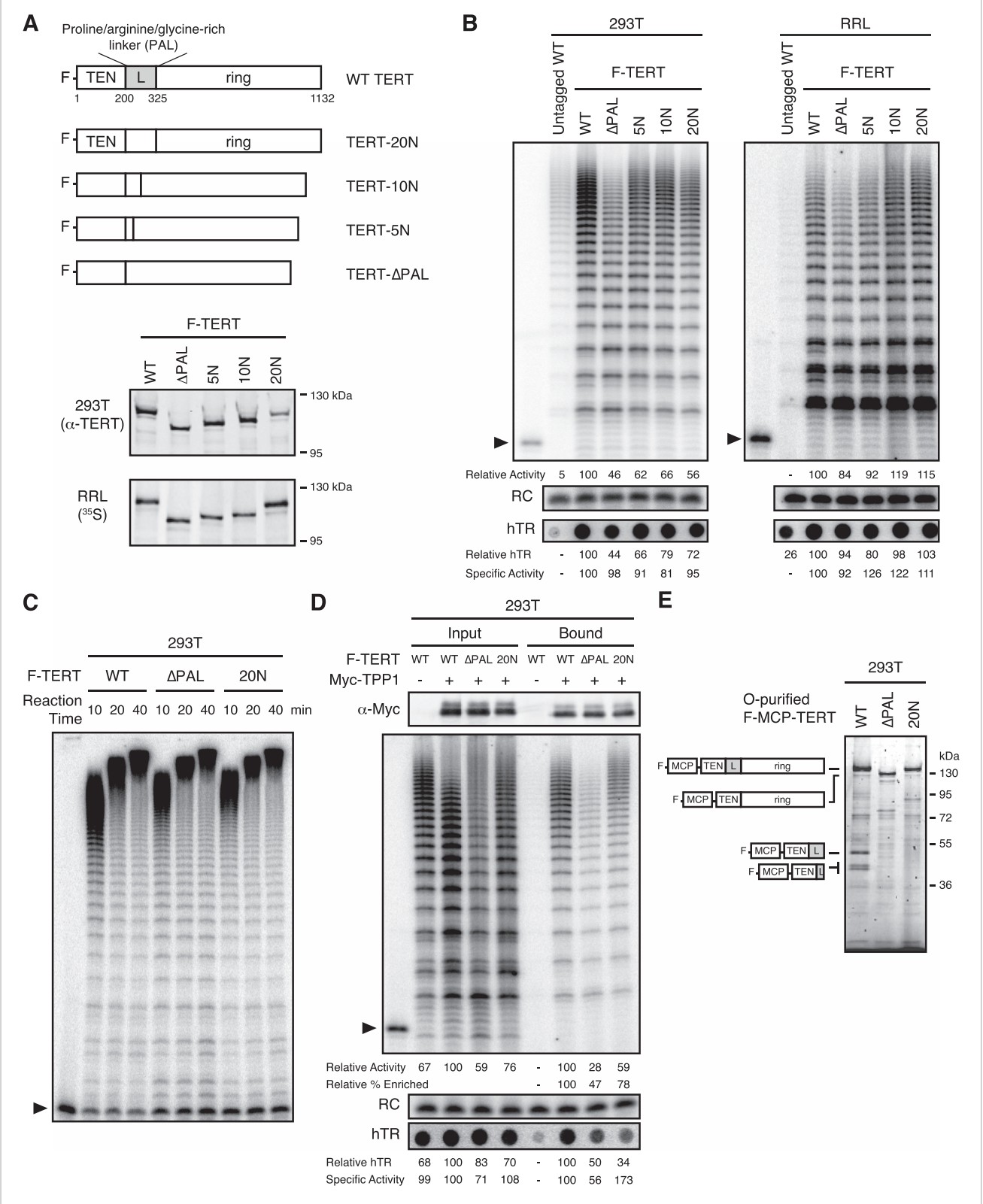

**Figure 6**. Telomerase RNP assembly and activity without the TEN domain linker. (**A**) Schematic representation and expression of N-terminally F-tagged human TERT proteins with the linker replaced by 20, 10, or 5 repeats of the sequence NAAIRS (TERT-20N through 5N) or linker deleted without compensating sequence insertion (TERT-ΔPAL). TERTs expressed in 293T cells were detected by immunoblot with TERT antibody, and TERTs expressed

*Figure 6. continued on next page*

*Figure 6. Continued*

in RRL were detected by $^{35}$S-methionine labeling during synthesis. (**B**) Activity and hTR content of 293T- or RRL-reconstituted, F-purified TERT RNPs with altered linker sequence, bound to FLAG antibody resin. Spot-blot hybridization was used to detect hTR. Relative activity and hTR content were normalized to the WT TERT purification after background subtraction of activity or hTR in the purification of untagged WT TERT. Specific activity was calculated from relative activity and relative hTR. (**C**) Processive extension of 5′-labeled $(T_2AG_3)_3$ primer by telomerase assembled with WT, ΔPAL, or 20N TERT bound to FLAG antibody resin. The labeled primer was extended for 5 min before chase addition of unlabeled primer for a total extension time of 10, 20, or 40 min. (**D**) Activity and hTR content of telomerase in 293T input extracts or bound to Myc antibody resin. TPP1 OB-fold domain expression and purification were confirmed by immunoblot detection of the 3xMyc tag. Immunoblot and activity assay with whole-cell extract used 2% of the total purification input. Half of the post-purification sample was used for activity assays and half for Myc immunoblot. Spot-blot hybridization was used to detect hTR. Relative activity and hTR content were normalized to the input or bound sample for TPP1 purification of WT TERT, after bound hTR background subtraction using the purification without tagged TPP1 OB-fold domain. Relative percentage enrichment was calculated as relative bound activity adjusted for relative input activity. Specific activity was calculated from relative activity and relative hTR. (**E**) SDS-PAGE analysis of O-purified 293T MCP-TERT complexes labeled using CoA-Cy5. MCP-TERT fragments resulting from proteolysis within the PAL of WT TERT are schematized, in comparison to full-length TERT.

profile the repeat addition processivity of the reconstituted enzymes, we assayed telomerase activity using a primer radiolabeled at its 5′ end rather than by extension with radiolabeled dNTPs. This also allowed the use of a non-limiting concentration of dGTP in the activity assay reaction (see 'Materials and methods'). A 5-min pulse of primer extension was followed by a chase period with excess unlabeled primer to eliminate telomerase reinitiation on released product DNA. Under these conditions, primer extension was highly processive for complexes of WT TERT, TERT-ΔPAL, and TERT-20N assembled in 293T cells (*Figure 6C*). Similar results were obtained with RRL-reconstituted enzymes (data not shown). We conclude that human TERT linker length and linker sequence have a very limited influence on the catalytic activity of reconstituted holoenzyme or minimal RNPs.

Telomerase-mediated telomere synthesis is strictly dependent on TEN domain interaction with the oligonucleotide/oligosaccharide-binding (OB) fold domain of the shelterin component TPP1 (*Xin et al., 2007*; *Schmidt et al., 2014*; *Sexton et al., 2014*). To test whether sequence substitutions of the PAL compromise catalytically active telomerase association with TPP1, we co-overexpressed N-terminally 3xMyc-tagged TPP1 OB-fold domain (TPP1 residues 88–249) with F-tagged WT TERT, TERT-ΔPAL, or TERT-20N in 293T cells. Within a twofold difference, the TPP1 OB-fold domain co-purified active telomerase regardless of TERT linker length or sequence (*Figure 6D*).

As additional characterization prior to single-molecule imaging, we analyzed Cy5-labeled O-purified MCP-tagged WT TERT, TERT-ΔPAL, and TERT-20N complexes by SDS-PAGE. Curiously, the 293T TERT-ΔPAL and TERT-20N bulk purification fractions lacked the TERT proteolysis products co-enriched by WT TERT (*Figure 6E*). The WT TERT proteolysis products correspond to the TEN domain fused to lengths of PAL ending at the two computationally identified low-complexity regions. A simple hypothesis to explain these findings is that an hTR-bound full-length TERT can co-purify a TERT fragment dimerized through the PAL. This would account for the detection of some PAL-containing TEN domain in the 293T-O bulk purification, which unlike the bulk F-purification should not directly enrich TERT fragments compromised for hTR binding. Bulk purification fractions of RRL-reconstituted WT TERT lacked the TERT proteolysis products detected in the 293T bulk purifications (*Figure 5B* and data not shown), suggesting that the TERT PAL may be a target of protease cleavage in cells. Furthermore, this proteolysis is specific for WT PAL sequence because no TEN domain fragments were observed the TERT-ΔPAL and TERT-20N purifications (*Figure 6E*). We note that although TERT proteolysis products are present in the 293T WT TERT bulk purifications, they may not be represented in the DNA-bound subset of TERT complexes assayed by single-molecule imaging.

## TERT dimer requirement for the PAL

To investigate the TERT subunit content of PAL-mutant TERT complexes bound to DNA, we first O-purified 293T- and RRL-reconstituted complexes of co-expressed ACP- and MCP-tagged WT TERT, TERT-ΔPAL, or TERT-20N. A dramatic decrease in TERT co-localization was observed for TERT-ΔPAL and TERT-20N relative to WT TERT (*Figure 7A*; 21% vs 5% co-localization in 293T samples, p = 0.0008, and 22% vs 2–3% in RRL samples, p < 0.0001). By calculations using a value of 82% TERT-labeling efficiency, RRL-reconstituted TERT-ΔPAL and TERT-20N complexes were 98% and 96% TERT monomer, respectively (*Figure 7B*). Even by modeling using the lower-bound underestimate of TERT-labeling efficiency, TERT

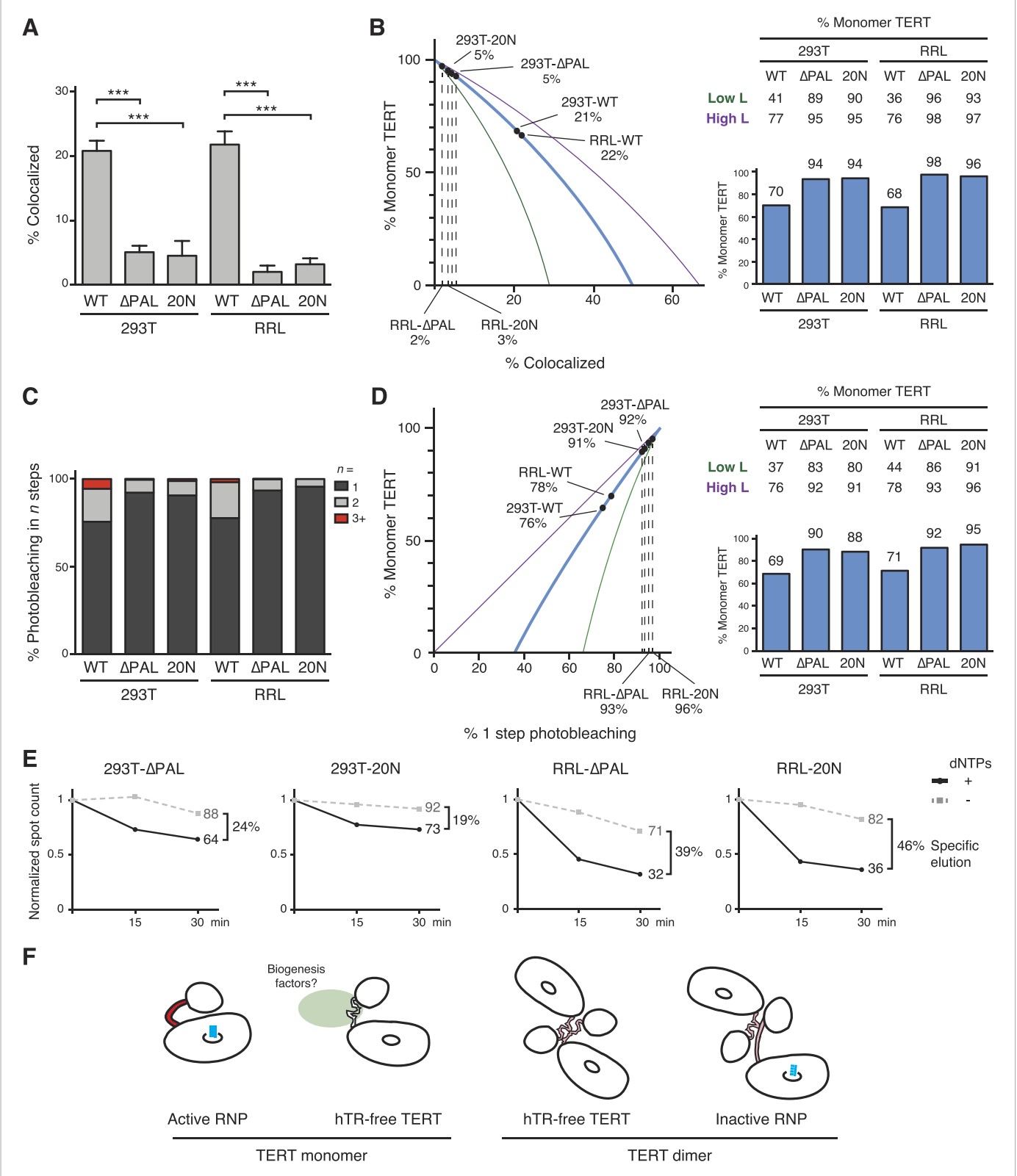

**Figure 7**. PAL-mediated TERT dimerization. (**A**) Two-color co-localization quantification for DNA-bound O-purified complexes of coexpressed ACP- and MCP-TERTs. Values are the average of triplicate experimental replicates. ***p < 0.001 using one-way ANOVA, followed by Tukey's multiple comparison test. (**B**) Calculated percentage of DNA-bound TERT monomer complexes according to the fraction of two-color TERT co-localization (percentages

*Figure 7. continued on next page*

*Figure 7. Continued*

indicated), assuming the TERT-labeling efficiency measured value (82%, blue line; bar graph at *right*), lower bound (51%, green line; Low L numbers at *right*), or upper bound (100%, purple line; High L numbers at *right*). (**C**) Photobleaching step quantification for DNA-bound O-purified MCP-TERT complexes labeled with Cy5. Values are the average of triplicate experimental replicates. (**D**) Calculated percentage of DNA-bound TERT monomer complexes according to the fraction of one-step photobleaching (percentages indicated), assuming the TERT-labeling efficiency measured value (82%, blue line; bar graph at *right*), lower bound (51%, green line; Low L numbers at *right*), or upper bound (100%, purple line; High L numbers at *right*). (**E**) Activity-dependent elution of O-purified Cy5-labeled MCP-TERT complexes using buffer containing dATP + dTTP or buffer only. Spot count per field of labeled TERT complexes was normalized to the initial time point of each sample. Specific elution was calculated by subtracting the fraction of complexes with buffer-only elution from the fraction eluted with dNTPs. (**F**) Illustration presenting the hypothesis of differences in TERT PAL conformation that occur with TERT RNP assembly or dimerization. The PAL is shown with conformations that correlate with catalytically active (red) or inactive (pink) TERT complexes.

monomer complexes were 93–96% of the DNA-bound RRL-reconstituted TERT complex total (*Figure 7B*). For 293T-reconstituted TERT-ΔPAL and TERT-20N complexes, TERT monomers were 94% of the population with a lower-bound underestimate of 89–90% (*Figure 7B*).

Parallel results were obtained by quantifying TERT subunit content using Cy5-labeled MCP-TERT steps of photobleaching. MCP-tagged TERT-ΔPAL and TERT-20N complexes assembled in 293T cells or RRL were dramatically depleted for multistep photobleaching compared to WT TERT complexes (*Figure 7C*; 24% vs 8–9% multistep bleaching in 293T samples, p = 0.0036, or 22% vs 4–7% in RRL samples, p = 0.0073). By calculations using a value of 82% TERT-labeling efficiency, RRL-reconstituted TERT-ΔPAL and TERT-20N complexes were 92% and 95% TERT monomer, respectively (*Figure 7D*; 86–96% across the modeled range of TERT-labeling efficiency). Similarly, 293T-reconstituted TERT-ΔPAL and TERT-20N complexes were 90% and 88% TERT monomer, respectively (*Figure 7D*; 80–92% across the modeled range of TERT-labeling efficiency).

To determine whether RNPs assembled with TERT-ΔPAL and TERT-20N retained the characteristic permutation dependence of human telomerase DNA binding, we tested Cy5-labeled MCP-tagged TERT-ΔPAL and TERT-20N complexes for activity-dependent elution. More elution of TERT-ΔPAL and TERT-20N complexes occurred in buffer + dNTPs than in buffer alone (*Figure 7E*). As observed for WT TERT complexes, the TERT-ΔPAL and TERT-20N complexes assembled in 293T cells showed lower efficiency of specific elution than complexes assembled in RRL. Nevertheless, specific elution of 293T and RRL complexes of TERT-ΔPAL or TERT-20N uniformly exceeded the fraction of TERT dimer complexes in each population, determined using subunit co-localization or photobleaching (*Figure 7B,D,E*).

We conclude that although PAL disruption drastically reduced TERT dimerization, RNPs assembled with PAL-mutant TERTs retained catalytic activity and even the permutation-dependent release of product DNA characteristic of the human telomerase active site. TERT dimerization was as effectively suppressed for telomerase holoenzyme assembled in cells as for minimal RNP assembled in RRL, suggesting that the TERT subunit content of reconstituted complexes has no dependence on any holoenzyme protein other than TERT. We speculate that in physiological context, protein interaction(s) mediated by the TERT PAL could chaperone hTR-free TERT from its synthesis in the cytoplasm to nuclear sites of RNP assembly (*Figure 7F*, left). TERT overexpression may bypass this chaperoning requirement and promote a TERT self-association disfavorable for TEN domain positioning relative to TERT ring in an active RNP (*Figure 7F*).

## Discussion

Understanding telomerase mechanism and regulation requires knowledge of the subunit stoichiometry of an active RNP. Whether assembled in vivo or in vitro, we show that human telomerase complexes monomeric for TERT are catalytically active. Monomeric TERT was abundant in the populations of DNA-bound complexes from at least three of the four bulk purification samples examined here, particularly from any purification using a template-complementary oligonucleotide. These results establish the phylogenetic conservation of a TERT-monomer telomerase active site. Also, our results support TERT haploinsufficiency rather than dominant-negative inhibition as the mechanism accounting for human disease from heterozygous TERT mutation (*Armanios et al., 2005*; *Armanios and Blackburn, 2012*). We note that although the budding yeast telomerase holoenzyme has a TERT monomer (*Bajon et al., 2015*), multiple telomerase RNPs can transiently co-localize as a

cluster (*Gallardo et al., 2011*). Similarly in human cells, Cajal bodies and/or shelterin interactions could dynamically cluster telomerase RNPs within a general nuclear area. The biological significance of this clustering remains to be determined (*Hockemeyer and Collins, 2015*).

The biased amino acid composition, low-complexity sequence, and predicted lack of structure of the TERT PAL all may promote overexpressed TERT formation of dimers and aggregates. The similar TERT monomer/dimer ratio observed for DNA-bound O-purified 293T vs RRL complexes suggests that TERT self-association accounts for the vast majority of dimer formation, since 293T and RRL TERT complexes differ in all components other than TERT (full-length hTR and H/ACA proteins vs hTRmin). TERT complexes with multistep photobleaching appeared to support little if any activity-dependent elution from DNA. It remains possible that active RNP dimers form under reconstitution conditions other than the standard protocols used in this work. Also, not all TERT monomer complexes had efficient activity-dependent elution: a larger fraction of 293T-O complexes than RRL-O complexes failed to elute with the opportunity for DNA synthesis, even when these complexes were converted to nearly homogeneous TERT monomer content by PAL deletion. We speculate that this difference arises from the greater heterogeneity of TERT structure, modification, and interaction partners produced by expression in cells. All of the findings above raise the need for caution in the interpretation of biochemical assays conducted using bulk purifications of TERT complexes. Surprisingly, even selection for single-stranded DNA-binding activity did not fully discriminate against inactive TERT.

We pinpoint a proline/arginine/glycine-rich human TERT domain linker as the major site of TERT dimerization. Although the PAL mediates dimerization of overexpressed TERT, at lower endogenous TERT expression level, we propose that the PAL has other biological roles. To address this hypothesis, it will be important to determine PAL interaction partners using approaches that recapitulate a physiological TERT expression level. Also, it will be of interest to understand which features of the TERT PAL are functionally significant. Because the PAL is present in vertebrate but not ciliate or budding yeast TERTs, we predict that it has biological function(s) related to the assembly of the vertebrate telomerase holoenzyme as an H/ACA RNP.

## Materials and methods

### Telomerase reconstitution in cells

HEK 293T cells were transiently transfected with pcDNA3.1 TERT expression plasmid(s), the hTR expression plasmid pBS-U3-hTR-500 (*Fu and Collins, 2003*), and where indicated, the N-terminally triple Myc-tagged TPP1 OB-fold domain (residues 88–249) expression plasmid pcDNA3.1-3xMyc-TPP1 (88–249) using calcium phosphate. After 48 hr, cells were resuspended in HLB buffer (20 mM HEPES at pH 8, 2 mM $MgCl_2$, 0.2 mM EGTA, 10% glycerol, 0.1% NP-40, 1 mM DTT, and 0.1 mM PMSF) and lysed by three freeze–thaw cycles. NaCl was adjusted to 400 mM and the whole-cell extract was cleared by centrifugation.

### Telomerase reconstitution in RRL

TNT T7 coupled transcription/translation reactions were assembled according to manufacturer's instructions (Promega, Madison, WI) with 40 ng/μl TERT expression plasmid and 100 ng/μl purified in vitro transcribed hTRmin added prior to TERT synthesis (*Wu and Collins, 2014a*). Reactions were incubated at 30°C for 3.5 hr.

### Enrichment of complexes by tagged TERT, tagged TPP1, or hTR template for activity assays

HEK 293T cell extracts (200 μl per precipitation) or RRL reconstitution reactions (37.5 μl per precipitation) were adjusted to 150 mM NaCl and bound to 10 μl FLAG M2 monoclonal antibody resin (Sigma–Aldrich, St. Louis, MO), 10 μl c-Myc antibody resin (Sigma–Aldrich) or 10 μl streptavidin agarose resin (Sigma–Aldrich) coated with 5′-biotinylated template-antisense oligonucleotide (CTAGACCTGTCATCAGUUAGGGUUAG, where the underlined nucleotides are 2′OMe RNA; [*Schnapp et al., 1998*]) by end-over-end rotation at room temperature for 2 hr. Following binding, the resin was washed three times at room temperature with HLB containing 150 mM NaCl, 0.1% Triton X-100, and 0.2% CHAPS. Resin-bound telomerase was then used in activity assay reactions (see below).

## Immunoblots

Immunoblotting for TERT detection was performed using mouse anti-TERT polyclonal primary antibody 1A4 raised against the TERT C-terminus at 1:3000 dilution. FLAG was detected using mouse anti-FLAG monoclonal primary antibody M2 (Sigma–Aldrich) at 1:5000 dilution. Tubulin was detected using mouse anti-alpha-tubulin monoclonal primary antibody DM1A (Calbiochem, Billerica, MA) at 1:500 dilution. Biotin was detected using goat anti-biotin polyclonal primary antibody ab6643 (Abcam, Cambridge, MA) at 1:5000 dilution. Myc was detected using rabbit anti-c-Myc polyclonal primary antibody A-14 (Santa Cruz Biotechnology, Dallas, TX) at 1:3000 dilution. Immunoblots using mouse primary antibodies were detected with goat anti-mouse IR 800 secondary antibody (Rockland Immunochemicals, Limerick, PA) at 1:20,000 dilution. Immunoblots using goat primary antibodies were detected with donkey anti-goat Alexa Fluor dye 680 secondary antibody (Life Technologies, Waltham, MA) at 1:15,000 dilution. Immunoblots using rabbit primary antibodies were detected with goat anti-rabbit IR 800 secondary antibody (Rockland Immunochemicals) at 1:20,000 dilution. All incubations were performed in 3% non-fat milk in Tris-buffered saline (TBS) buffer (50 mM Tris pH 7.5, 150 mM NaCl). Membranes were washed with TBS buffer prior to visualization on a LI-COR Odyssey imager (LI-COR Biotechnology, Lincoln, NE).

## Telomerase activity assays

Primer extension assays with radiolabeled nucleotide incorporation were performed in 20 µl reactions containing 10 µl resin-bound telomerase, 500 nM $(T_2AG_3)_3$ telomeric primer, and >0.1 µM $\alpha$-$^{32}$P dGTP (3000 Ci/mmol, 10 mCi/ml, Perkin–Elmer, Waltham, MA) in telomerase activity assay buffer (50 mM Tris-acetate at pH 8, 3 mM $MgCl_2$, 1 mM EGTA, 1 mM spermidine, 5 mM DTT, and 5% glycerol) with 5 µM dGTP, 250 µM dTTP, and dATP for detection of repeat addition processive synthesis or 250 µM dTTP and 500 µM ddATP for detection of single-repeat synthesis. Reactions were incubated at 30°C for 40 min. For the 5′-end labeled primer extension pulse-chase assay, 10 µl resin-bound telomerase was incubated with 20 nM $^{32}$P 5′-end labeled $(T_2AG_3)_3$ telomeric primer for 30 min, then washed twice with HLB containing 150 mM NaCl and 0.1% NP-40 to remove unbound primer. The assay was initiated by addition of 20 µl of telomerase activity assay buffer with 250 µM dGTP, dTTP, and dATP. The reactions were incubated at 30°C for 5 min followed by addition of unlabeled $(T_2AG_3)_3$ telomeric primer to a final concentration of 5 µM and further incubated at 30°C to reach the indicated total reaction time.

The products of all activity assay reactions were then extracted, precipitated, and resolved on 12% polyacrylamide/7 M urea/0.6× Tris borate-EDTA gels. An end-labeled oligonucleotide was added prior to product precipitation to serve as a recovery control, and end-radiolabeled primer was loaded separately from product DNA as a size marker (migration is indicated in Figures by ►). Dried gels were visualized by phosphorimaging on a Typhoon Trio system (GE Healthcare, Piscataway, NJ) and quantified using ImageQuant TL (GE Healthcare). Activity was quantified on the combined intensities of all product DNA.

## ACP/MCP labeling with CoA derivatives

Complexes bound to an affinity purification resin were washed into 50 mM HEPES at pH 8, 1 mM DTT, and 10 mM $MgCl_2$. CoA-conjugated biotin was purchased (New England Biolabs, Ipswich, MA) and CoA-conjugated Cy3 or Cy5 was prepared as described (Yin et al., 2006) and added to a final concentration of 10 µM. Labeling reactions were carried out by addition of ACP or SFP synthase (New England Biolabs) to 1 µM final concentration and incubation at room temperature for 2 hr. Following the labeling reaction, the resin was washed three times at room temperature with HLB containing 150 mM NaCl, 0.1% Triton X-100, and 0.2% CHAPS. For samples sequentially labeled with two dyes, the labeling reactions were repeated with the second dye and synthase. After the final labeling reaction and wash, complexes were eluted by incubation with 200 nM FLAG peptide or 30 µM 3′-terminal 2′,3′-dideoxyguanosine-modified displacement oligonucleotide (CTAACCCTAACTGAT-GACAGGTCTAG; [Schnapp et al., 1998]) for 1 hr at room temperature. Complexes bound to FLAG antibody or 2′OMe RNA oligonucleotide resin were eluted in 14 µl or 70 µl buffer, respectively. These volumes were required to normalize activity and fluorescent spot count among preparations from the same amount of input. Labeled bulk samples were analyzed by 10% SDS-PAGE and imaged on a Typhoon Trio system (GE Healthcare).

## Labeled TERT depletion

Telomerase was reconstituted with ACP- and/or MCP-TERT as described above, with the RRL reaction supplemented with $^{35}$S-methionine. Following FLAG purification, complexes were labeled with CoA or CoA-biotin with ACP or SFP synthase. Samples were eluted from the affinity purification resin with 200 nM FLAG peptide and bound to streptavidin agarose or Myc antibody agarose (Sigma–Aldrich) for 1 hr. For depletion in denaturing conditions, samples were eluted from affinity purification resin in buffer adjusted to 2 M urea. The streptavidin-agarose unbound fraction was analyzed by 10% SDS-PAGE or activity assay.

## Microscopy

A prism-type total internal reflection fluorescence microscope was built using a Nikon Ti-E Eclipse inverted fluorescence microscope equipped with a 60× 1.20 N.A. Plan Apo water objective (Nikon Instruments, Melville, NY). A 532-nm laser (Coherent, Inc., Santa Clara, CA, 350 mW) was used for Cy3 excitation, and a 633-nm laser (JDSU, Milpitas, CA, 35 mW) was used for Cy5 excitation. For two-color co-localization experiments, Cy3 and Cy5 fluorescence were split into two channels and imaged separately on a single charge-coupled device (CCD) chip using an Optosplit II image splitter (Cairn Instruments, Faversham, UK). Fluorescence signal was collected with a 512 × 512 pixel electron-multiplied CCD camera (Andor Technology, Belfast, UK). All data collection was conducted at 22°C.

## Slide preparation

Quartz coverslips were coated with a mixture of 99% PEG and 1% biotinylated-PEG. Airtight sample chambers were constructed by sandwiching double-sided tape between the coverslips and quartz slides (MicroSurfaces, Inc., Englewood, NJ). To prepare the slides for molecule deposition, the surface was pre-blocked by sequential 15-min incubations with 20% Biolipidure 203/206 (NOF Corporation, White Plains, NY) and 10 mg/ml casein (Sigma–Aldrich). Following each incubation, the sample chamber was washed with telomerase slide buffer (50 mM Tris-HCl at pH 8, 10% glycerol, 2 mM MgCl$_2$, and 0.2 mM EGTA). The surface was then incubated with 1 mg/ml streptavidin (Sigma–Aldrich) for 10 min and washed twice with telomerase slide buffer.

## Two-color co-localization and photobleaching analyses

Streptavidin-coated slides were incubated with 40 nM 5′-biotinylated telomeric primer (Tel2, T$_{15}$TTAGGGTTAGGG) in telomerase activity assay buffer for 10 min and washed with telomerase slide buffer. The slide was then incubated for 30 min with 1 µl labeled telomerase supplemented with 1 mg/ml casein followed by two washes with telomerase slide buffer to remove excess unbound sample. After washing, imaging buffer (1 mg/ml glucose oxidase, 0.34 mg/ml catalase, 0.8% wt/vol D-glucose, and 2 mM Trolox in telomerase slide buffer) was flowed into the sample chamber.

The fraction of two-color co-localization was experimentally determined considering only complexes with Cy5 signal and measuring the percentage of the spots that also had Cy3 signal. This was done because initial Cy5 labeling of the ACP tag by ACP synthase is selective for ACP vs MCP tag, whereas the subsequent SFP synthase labeling used to add Cy3 can label both MCP and ACP tags. By only considering complexes that labeled with Cy5, we avoided the possibility of counting two-TERT single-color Cy3 labeled complexes as TERT monomers rather than dimers. Samples were excited with the 633-nm laser throughout the experiment and imaged at 100-ms time resolution. After the first 10–20 frames, samples were excited with the 532-nm laser for ~20 additional frames. For photobleaching, the 633-nm laser was used for excitation and 500–1000 frames were collected at 100-ms time resolution.

## Activity-dependent elution

Tel2-bound slides were incubated with 1 µl Cy5-labeled telomerase in telomerase activity assay buffer for 30 min, and then washed twice. Antisense Tel2 oligonucleotide (Anti-Tel2, CCCTAACCCTAA) was then introduced at 100 nM final concentration and incubated for 15 min to block any unbound immobilized Tel2. The slide was washed twice, and imaging buffer was flowed into the sample chamber. The samples were excited at 633 nm to collect 30 frames at 100-ms time resolution to determine the initial number of complexes bound to immobilized Tel2. For assays of elution, after initial imaging, the slide was washed and incubated with either 20 µl dNTP elution buffer (10 nM Anti-Tel2, 500 µM dATP, and 500 µM dTTP in telomerase activity assay buffer) or mock elution buffer

(10 nM Anti-Tel2 in telomerase activity assay buffer). After 15 and 30 min, the slide was then washed with telomerase activity assay buffer, imaging buffer was flowed into the imaging chamber, and the remaining number of bound complexes was determined by collecting 30 frames at 100-ms time resolution with 633-nm excitation. For photobleaching step quantification after elution, no initial imaging or imaging at 15 min was performed. For quantification of RNase sensitivity, Cy5-labeled MCP-TERT reconstitutions were pre-incubated with 0.1 mg/ml RNase A at room temperature for 1 hr immediately prior to introduction to the flow cell.

## Determination of DNA $K_d$

Streptavidin-coated slides were incubated with 1 μl biotin-labeled sample diluted in telomerase activity assay buffer for 10 min. The sample chamber was washed with telomerase slide buffer and incubated with 500 nM non-specific blocking oligonucleotide (AAATGATAACCATCTCGC) for 15 min, followed by two washes with telomerase slide buffer. Telomeric oligonucleotide (TTAGGGTTAGGG) 5′-end labeled with Cy5 was incubated for 15 min. Excess DNA was washed away and imaging buffer was flowed into the sample chamber. Bound DNA was detected by collecting 30 frames at 100-ms time resolution with 633-nm excitation.

## Northern blots

RNA was purified using TRIzol reagent (Life Technologies) and resuspended in 2 μl of water. The RNA was spotted onto Hybond N+ nylon membrane (GE Healthcare) and detected using $^{32}$P end-labeled probe complementary to hTR positions 51–72 (*Fu and Collins, 2003*).

## Equations

### Equation 1

Here, we derive an expression for the probability of a complex containing one TERT molecule as a function of labeling efficiency and measured two-color colocalization. Below 'monomer' and 'dimer' are used to indicate TERT subunits within complexes.

We model the telomerase complexes as having either one or two TERT molecules.

Let $M$ = probability of monomer and $D$ = probability of dimer:

$$M + D = 1.$$

Therefore,

$$D = 1 - M.$$

The probability of a monomeric complex containing ACP- or MCP-TERT is assumed to be equivalent. Furthermore, we considered the ACP and MCP tags to label at the same efficiency L. Therefore, the probability of a monomeric TERT complex labeled with Cy5 (Red, $R$) or Cy3 (Green, $G$) expressed as a function of $M$ and $L$ is:

$$P(R) = P(G) = 0.5ML.$$

Complexes containing two TERTs could have two copies of ACP-TERT or two copies of MCP-TERT (denoted as *same*) or one of each (denoted as *mixed*). The ACP$_{same}$, MCP$_{same}$ and ACP/MCP$_{mixed}$ populations exist in a ratio of 0.25:0.25:0.5, respectively. Below, the ACP tag labeled with Cy5 is denoted as $R$ and the MCP tag labeled with Cy3 is denoted as $G$.

Considering complexes with two copies of ACP-TERT or two copies of MCP-TERT, the probability that both subunits are labeled is:

$$P(RR) = P(GG) = 0.25DL^2 = 0.25(1-M)L^2.$$

The probability that one of the subunits is labeled while the other is unlabeled (0) is:

$$P(R0_{same}) = P(0R_{same}) = P(G0_{same}) = P(0G_{same}),$$

$$= 0.25D(1-L)L = 0.25(1-M)(1-L)L.$$

Complexes with one copy of ACP-TERT and one copy of MCP-TERT can be labeled on both subunits:

$$P(RG|GR) = 0.5DL^2 = 0.5(1-M)L^2,$$

or labeled on one subunit only:

$$P(R0|0R_{mixed}) = P(G0|0G_{mixed}) = 0.5D(1-L)L = 0.5(1-M)(1-L)L.$$

The fraction of colocalization was experimentally determined considering only complexes with Cy5 signal ($R$) and measuring the percentage of the spots that also had Cy3 signal ($G$). This was done because initial Cy5 labeling of the ACP tag by ACP synthase is selective for ACP versus MCP tag, whereas the subsequent SFP synthase labeling used to add Cy3 can label both MCP and ACP tags. By only considering complexes that labeled with Cy5, we avoided the possibility of counting two-subunit single-color Cy3 labeled complexes as monomers rather than dimers.

The probability of colocalization, $C$, is therefore the probability of a dimer with one Cy5-labeled subunit and one Cy3-labeled subunit ($RG|GR$) normalized to all complexes with a Cy5-labeled subunit (*Any R*).

$$C = \frac{P(RG|GR)}{P(Any\ R)} = \frac{P(RG|GR)}{\sum P(R), P(RG|GR), P(R0_{same}), P(0R_{same}), P(R0|0R_{mixed}), P(RR)},$$

$$= \frac{0.5(1-M)L^2}{0.5ML + 0.5(1-M)L^2 + 0.25(1-M)(1-L)L + 0.25(1-M)(1-L)L + 0.5(1-M)(1-L)L + 0.25(1-M)L^2},$$

$$= \frac{-2(M-1)L}{M(L-2)-L+4}.$$

Solving for $M$:

$$M = \frac{L(C+2)-4C}{L(C+2)-2C}.$$

## Equation 2

Here, we derive an expression for the probability of a complex containing one labeled TERT as a function of labeling efficiency and measured one-step photobleaching.

We model the telomerase complexes as having either one or two TERT molecules.

Let $M$ = probability of monomer and $D$ = probability of dimer:

$$M + D = 1.$$

Therefore,

$$D = 1 - M.$$

For photobleaching experiments, telomerase was reconstituted with MCP-TERT labeled with Cy5. The probability of a monomeric complex with Cy5 (Red, $R$) in terms of the labeling efficiency $L$ is:

$$P(R) = ML.$$

Complexes with two TERT molecules could have one or both subunits labeled. The probability that both subunits are labeled is:

$$P(RR) = DL^2 = (1-M)L^2.$$

The probability that one subunit is labeled while the other is unlabeled (0) is:

$$P(R0) = P(0R) = D(1-L)L = (1-M)(1-L)L.$$

The probability of one-step photobleaching, $B_1$, as a function of $M$ and $L$ is the probability of any complex with exactly one Cy5-labeled subunit normalized to all complexes with a Cy5-labeled subunit (*Any R*):

$$B_1 = \frac{\sum P(R), P(R0), P(0R)}{P(Any\ R)} = \frac{\sum P(R), P(R0), P(0R)}{\sum P(R),\ P(RR), P(R0), P(0R)},$$

$$= \frac{ML + (1-M)(1-L)L + (1-M)(1-L)L}{ML + (1-M)L^2 + (1-M)(1-L)L + (1-M)(1-L)L},$$

$$= \frac{2ML - 2L - M + 2}{ML - L - M + 2}.$$

Solving for $M$:

$$M = \frac{L(B_1 - 2) - 2B_1 + 2}{L(B_1 - 2) - B_1 + 1}.$$

## Equation 3

Here, we determine the probability of two-step photobleaching as a function of the fraction of colocalization. This gives an indication for how cross-consistent colocalization and multistep bleaching results are with each other.

Combining Equation 1, where $C$ = probability of colocalization:

$$M = \frac{L(C + 2) - 4C}{L(C + 2) - 2C},$$

and Equation 2, where $B_1$ = probability of one-step photobleaching:

$$M = \frac{L(B_1 - 2) - 2B_1 + 2}{L(B_1 - 2) - B_1 + 1},$$

it follows that:

$$\frac{L(C + 2) - 4C}{L(C + 2) - 2C} = \frac{L(B_1 - 2) - 2B_1 + 2}{L(B_1 - 2) - B_1 + 1}.$$

Therefore, the probability of one-step photobleaching as a function of the fraction of two-color colocalization is:

$$B_1 = \frac{3C - 2}{C - 2}.$$

Since it is assumed that the telomerase complexes have either one or two TERT molecules, all labeled complexes should photobleach in either one or two steps. Let $B_2$ = probability of two-step photobleaching:

$$B_1 + B_2 = 1.$$

Therefore,

$$B_2 = 1 - B_1.$$

The probability of two-step photobleaching as a function of the fraction of two-color colocalization is:

$$B_2 = 1 - B_1 = 1 - \frac{3C - 2}{C - 2},$$

$$B_2 = \frac{-2C}{C - 2}.$$

## Acknowledgements

We thank Aaron Whiteley for assistance with purification of CoA conjugates and Dirk Hockemeyer and Collins lab members for constructive discussions.

# Additional information

## Funding

| Funder | Grant reference | Author |
| --- | --- | --- |
| National Heart, Lung, and Blood Institute | HL079585 | Robert Alexander Wu, Kathleen Collins |
| National Institute of General Medical Sciences | GM094522, GM054198 | Robert Alexander Wu, Yavuz S Dagdas, S Tunc Yilmaz, Ahmet Yildiz, Kathleen Collins |
| National Science Foundation | DGE-1106400, MCB-1055017 | Robert Alexander Wu, Yavuz S Dagdas, Ahmet Yildiz |

The funders had no role in study design, data collection and interpretation, or the decision to submit the work for publication.

## Author contributions

RAW, Conception and design, Acquisition of data, Analysis and interpretation of data, Drafting or revising the article; YSD, Performed and analyzed single-molecule imaging experiments, Acquisition of data, Analysis and interpretation of data, Drafting or revising the article; STY, Performed and analyzed single-molecule imaging experiments; AY, Analysis and interpretation of data, Drafting or revising the article; KC, Conception and design, Analysis and interpretation of data, Drafting or revising the article

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
