## [Decision Letter]

Thank you for submitting your work entitled “Single-molecule imaging of telomerase reverse transcriptase in human telomerase holoenzyme and minimal RNP complexes” for peer review at *eLife*. Your submission has been favorably evaluated by James Manley (Senior Editor) and three reviewers, one of whom is a member of our Board of Reviewing Editors.

The reviewers have discussed the reviews with one another and the Reviewing Editor has drafted this decision to help you prepare a revised submission.

This manuscript addresses a long-standing debate over the subunit composition of human telomerase and whether the enzyme functions as a monomer or dimer. Resolution of this question is fundamental to full understanding of telomerase mechanism and how heterozygous TERT or telomerase RNA mutations cause human disease. The current manuscript circumvents the past problems associated with bulk population assays by taking very powerful and creative single molecule approaches to analyze both subunit composition and enzyme activity. They show unequivocally that telomerase is fully functional as a monomer: i.e. it shows the expected high affinity primer binding and activity with high repeat addition processivity. They also show that the approach used for enzyme production and purification has major effects on the fraction of enzyme in monomeric versus multimeric form.

Essential revisions:

1) In Figure 1, the western blots for TERT are done using an antibody that has not been characterized – the Methods simply cite “Geron”. Given the significant literature about the non-specificity of TERT antibodies, it is important to show the specificity of the antibody used here.

2) In Figure 1, two different kinds of telomerase activity assays are shown in E and F, but there is no comment about these two different methods in the Methods section.

3) In the experiments presented in Figure 4, it is not clear why the authors don't look at the fraction of co-localized spots AFTER product release. If the monomers are more specifically released, there should be an increase in co-localization after release. This is a very straightforward experiment that would extend the findings.

4) In Figure 6, the data on the TEN swapping experiments are confusing and should be removed from the manuscript. At the end of the subsection “Assessing the active RNP fraction of DNA-bound TERT complexes”, the authors conclude that the human TERT is a monomer. Then, in the next subsection (“TERT dimerization without TERT or hTR domain swapping”), the authors start by saying: “we next addressed the biochemical basis for assembly of two TERT subunits”. This makes no sense. Why would you do these experiments after having shown the active enzyme is a monomer? The next section on the PAL domain should be included but should be set up much more clearly as a way to investigate how TERT might be found in aggregates when the active enzyme is a monomer. The authors should seriously consider removing this data and focusing the paper.

5) In Figure 7, the data in panel C is messy and it is not clear how this adds to the story. The limited proteolysis of an isolated domain is not very helpful. If the authors showed that full length TERT is proteolyzed within the PAL region that would be a much better use of this technique. As it is, this panel does not add to the paper.

6) Again in Figure 7, the authors conclude that there is no effect of the PAL deletion on telomerase activity, but from the gels in Figure 7 there is a visible effect. Showing reproducible side-by-side lane would be helpful here, and more importantly a dilution series of the elongation products, or a time course to better compare activity is needed. The difference seen in these gels is almost as great as in Figure 3 where the authors state a 50% reduction in activity.

---

## [Author Response]

*1) In*
Figure 1*, the western blots for TERT are done using an antibody that has not been characterized – the Methods simply cite “Geron”. Given the significant literature about the non-specificity of TERT antibodies, it is important to show the specificity of the antibody used here*.

Panel D has been added to Figure 1—figure supplement 1 to demonstrate the specificity of this antibody.

*2) In*
Figure 1*, two different kinds of telomerase activity assays are shown in E and F, but there is no comment about these two different methods in the Methods section*.

The text, Methods and figure legends now detail the two activity assay conditions, which differ in whether repeat synthesis was constrained to a single telomeric repeat or allowed to be processive. The former was optimal for quantifying relative amounts of enzyme whereas the latter was necessary for comparing the overall profiles of product synthesis.

*3) In the experiments presented in*
Figure 4*, it is not clear why the authors don't look at the fraction of co-localized spots AFTER product release. If the monomers are more specifically released, there should be an increase in co-localization after release. This is a very straightforward experiment that would extend the findings*.

Panels D, E and F have been added to revised Figure 4 to address this comment, which relates to the imaging of Cy5-labeled MCP-TERT complexes over a time course of activity-dependent DNA release. TERT subunit counting in an MCP-TERT reconstitution would be by steps of photobleaching rather than two-color colocalization, but the intent of this request is clear. We had hesitated to include quantification of photobleaching steps on a post-elution field of TERT complexes already imaged several times to quantify spot count per microscope field. We resolved this concern by performing comparative post-elution analysis of photobleaching as an experiment separate from the spot-count time courses described by Figure 4 panels B, C. The new results in Figure 4 confirm the prediction that “the monomers are more specifically released” by product synthesis.

*4) In*
Figure 6*, the data on the TEN swapping experiments are confusing and should be removed from the manuscript. At the end of the subsection “Assessing the active RNP fraction of DNA-bound TERT complexes”, the authors conclude that the human TERT is a monomer. Then, in the next subsection (“TERT dimerization without TERT or hTR domain swapping”), the authors start by saying: “we next addressed the biochemical basis for assembly of two TERT subunits”. This makes no sense. Why would you do these experiments after having shown the active enzyme is a monomer? The next section on the PAL domain should be included but should be set up much more clearly as a way to investigate how TERT might be found in aggregates when the active enzyme is a monomer. The authors should seriously consider removing this data and focusing the paper*.

All of the domain-swapping experiments and all of the isolated TEN domain reconstitution experiments have been removed from the revised manuscript.

*5) In*
Figure 7*, the data in panel C is messy and it is not clear how this adds to the story. The limited proteolysis of an isolated domain is not very helpful. If the authors showed that full length TERT is proteolyzed within the PAL region that would be a much better use of this technique. As it is, this panel does not add to the paper*.

All experiments characterizing the TEN domain in isolation have been removed from the revised manuscript.

*6) Again in*
Figure 7*, the authors conclude that there is no effect of the PAL deletion on telomerase activity, but from the gels in*
Figure 7
*there is a visible effect. Showing reproducible side-by-side lane would be helpful here, and more importantly a dilution series of the elongation products, or a time course to better compare activity is needed. The difference seen in these gels is almost as great as in*
Figure 3
*where the authors state a 50% reduction in activity*.

The new Figure 6 expands the analysis of the influence of PAL deletion on telomerase activity previously presented in Figure 7. New experiments include side-by-side quantifications of RNP assembly efficiency and specific activity and also the requested time course analysis of repeat addition processivity.